# Layer by Layer: Uncovering Hidden Representations in Language Models

Oscar Skean [1]   Md Rifat Arefin [2 3]   Dan Zhao [4]   Niket Patel [5]   Jalal Naghiyev [6]
Yann LeCun [4 7]   Ravid Shwartz-Ziv [4 8]

## Abstract

From extracting features to generating text, the outputs of large language models (LLMs) typically rely on the final layers, following the conventional wisdom that earlier layers capture only low-level cues. However, our analysis shows that *intermediate layers* can encode even richer representations, often improving performance on a range of downstream tasks. To explain and quantify these hidden-layer properties, we propose a unified framework of representation quality metrics based on information theory, geometry, and invariance to input perturbations. Our framework highlights how each layer balances information compression and signal preservation, revealing *why* mid-depth embeddings can exceed the last layer's performance. Through extensive experiments on 32 text-embedding tasks across various architectures (transformers, state-space models) and domains (language, vision), we demonstrate that intermediate layers consistently provide stronger features, challenging the standard view on final-layer embeddings and opening new directions on using mid-layer representations for more robust and accurate representations.

## 1. Introduction

Large Language Models (LLMs) have driven remarkable progress in natural language processing (NLP), achieving state-of-the-art results on many tasks (Brown et al., 2020; Devlin et al., 2019; Li et al., 2022). At the heart of most applications lies a common assumption: *final-layer representations* are the most useful for downstream tasks. Yet a fundamental question remains: *does the final layer always yield the best representation?*

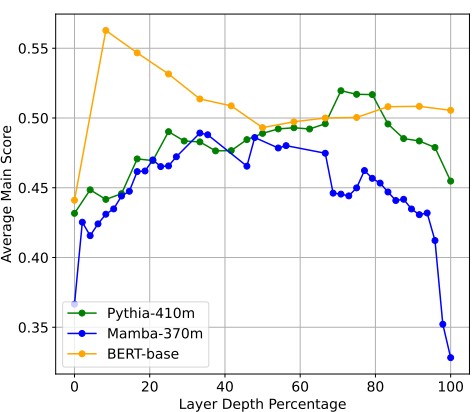

Figure 1: **Intermediate layers consistently outperform final layers on downstream tasks.** The average score of 32 MTEB tasks using the outputs of every model layer as embeddings for three different model architectures. The x-axis is the depth percentage of the layer, rather than the layer number which varies across models.

In this paper, we conduct a *layer-wise* analysis of LLMs across diverse architectures—including transformer-based ones (Vaswani et al., 2017), state-space models (SSMs) (Gu & Dao, 2024), and encoder-based models like BERT (Devlin et al., 2019)—spanning parameter scales from tens of millions to billions. Through systematic evaluation on 32 embedding tasks from the **Massive Text Embedding Benchmark (MTEB)** (Muennighoff et al., 2022), we find that *intermediate layers* often surpass the final layer by up to 16% in downstream accuracy. Figure 1 illustrates this phenomenon, where mid-depth layers provide particularly strong representations while the very last layer can become overly specialized to the pretraining objective.

**A unified framework.** To better understand intermediate layers' effectiveness, we combine three complementary perspectives (Section 3):

- **Information-theoretic:** How much do layers compress or preserve semantic information (Shwartz-Ziv & Tishby, 2019; Shwartz-Ziv, 2022)?

- **Geometric:** How do token embeddings unfold in high-dimensional space (Hosseini & Fedorenko, 2023))?

[1]University of Kentucky [2]Mila [3]University of Montreal [4]New York University [5]University of California, Los Angeles [6]Independent [7]Meta FAIR [8]Wand.AI. Correspondence to: Oscar Skean <oscar.skean@uky.edu>.

*Proceedings of the 42nd International Conference on Machine Learning*, Vancouver, Canada. PMLR 267, 2025. Copyright 2025 by the author(s).

- **Invariance:** Are embeddings robust to input perturbations (e.g., InfoNCE (Oord et al., 2018), LiDAR (Thilak et al., 2024) and DiME (Skean et al., 2023))?

We show that these perspectives can be viewed under a single lens, which clarifies how intermediate layers strike a balance between retaining features and discarding noise.

**Key findings and contributions.** Our investigation leads to several important insights:

- *Intermediate layers consistently outperform final layers.* This pattern is evident in both transformers and SSMs, suggesting a broad architecture-agnostic effect.

- *Autoregressive vs. masked-language training.* Autoregressive models exhibit a pronounced mid-layer "compression valley," whereas masked or bidirectional models show milder intermediate changes.

- *Domain-general effect.* We extend these results to vision models and find that autoregressive image transformers display the same mid-depth bottleneck, indicating that the *training objective*, rather than the data modality, is the key driver.

- *CoT finetuning.* Analyzing chain-of-thought (CoT) reveals that finetuning can reshape mid-layer entropy, preserving latent context for multi-step reasoning.

Overall, our results challenge the default reliance on final-layer embeddings and highlight intermediate layers as potentially underutilized sources of meaningful features. In this paper, we detail our unified framework (Section 3), present extensive experiments in both language and vision (Section 4, 5, 6), and conclude with a discussion of our findings, their implications, and future directions. [1]

## 2. Related Work

**Understanding Neural Representations.** A long line of research has aimed to understand *how* deep neural networks encode and organize information. Early studies employed linear probes for intermediate layers (Alain & Bengio, 2017), while subsequent efforts introduced more sophisticated techniques such as SVCCA (Raghu et al., 2017) to compare learned features across architectures and training regimes. Although these approaches have shed light on representation dynamics, most focus on vision models or shallow networks. In contrast, our work contributes to a growing body of literature extending layer-wise analysis to *large-scale* language models, emphasizing specific behaviors of intermediate layers across diverse architectures. Complementing our empirical findings, Saponati et al. (2025) present a theoretical analysis of how different pretext

tasks, such as next-token prediction and masked language modeling, influence the structure of learned representations.

**Layer-wise Analysis in Language Models.** Recent work has increasingly focused on identifying *which* transformer layers encode different types of information. For example, linguistic features such as part-of-speech tags or semantic roles are best encoded by the middle layers of a BERT (Liu et al., 2019; Tenney et al., 2019; Voita et al., 2019). More recent work has shown that mid-depth layers sometimes hold surprisingly robust features, challenging the usual emphasis on final layer representations (Jin et al., 2024; Gurnee & Tegmark, 2023; Fan et al., 2024). A related line of work investigates the attention sink phenomenon (Xiao et al., 2024; Brunner et al., 2020; Gu et al., 2025), in which attention disproportionately concentrates on a single token. Notably, intermediate decoder layers have been shown to not exhibit these extreme attention sinks (Barbero et al., 2025), suggesting they engage in more distributed and meaningful information processing than the shallow or deep layers.

**Compression and Generalization.** Multiple lines of research link compression and generalization performance (Deletang et al., 2024). For instance, Bordes et al. (2023) demonstrated that discarding certain layers in self-supervised encoders can even *improve* downstream accuracy, while Park et al. (2024a) found that LLM embeddings often lie in low-dimensional manifolds. Our empirical study reinforces these ideas by demonstrating that many networks—especially autoregressive transformers—naturally develop a mid-layer bottleneck that appears crucial for balancing "signal" versus "noise." We show how intermediate layers can achieve optimal trade-offs between preserving task-relevant information and discarding superfluous detail.

**Representation Quality Metrics.** A variety of metrics have been proposed to quantify the "quality" of learned representations. We group them into three main categories:

- **Information-theoretic measures** capture how much a model's internal representations compress or preserve relevant information. For example, the Information Bottleneck (Shwartz-Ziv & Tishby, 2019; Shwartz-Ziv, 2022) analyzes whether intermediate layers discard noise while retaining essential features. Intrinsic dimensionality, which describes the minimum number of features to represent data, has also been used to analyze intermediate layers in LLMs (Cheng et al., 2025; Valeriani et al., 2023; Razzhigaev et al., 2024). This line of work has shown semantic abstractions useful for downstream tasks are better encoded in middle layers than last layers in large transformer models. While we do not study intrinsic dimensionality in our subsequent analysis, it would make a promising direction

---

[1]We make our code available at `https://github.com/OFSkean/information_flow`

for future work.

- **Geometric measures** focus on the structure of embeddings in high-dimensional space. Classical approaches include analyzing singular values and effective rank of the representation matrix (Garrido et al., 2023). The anisotropy metric of Razzhigaev et al. (2024) has been used to study compression in intermediate model layers and we compare our results with their findings in Section 4.2. Anisotropy fits in well with our proposed framework, though we leave a formal integration to future work. Recent work explores curvature (Hosseini & Fedorenko, 2023) to quantify how smoothly tokens are mapped across consecutive positions or time steps.

- **Task-based or invariance metrics** evaluate how well representations support downstream goals. For instance, augmentations-based approaches such as InfoNCE (Oord et al., 2018) and LiDAR (Thilak et al., 2024) estimate invariance to perturbations, while methods like NESum or Self-Cluster (Agrawal et al., 2022) link closely to entropy. In computer vision, these scores often correlate strongly with downstream accuracy, highlightFing how robust the embeddings are.

Although these representation quality metric categories may appear distinct, we show (Section 3) that many can be unified under a single lens. This unification illuminates *why* certain intermediate layers balance compression, geometry, and invariance so effectively, leading to better representations for downstream tasks.

Overall, our work bridges these overlapping threads by evaluating a range of architectures and training paradigms via a unified set of metrics. Beyond merely confirming that intermediate layers can be effective, we elucidate *why* this happens, tying it to fundamental properties such as entropy, invariance, and geometry. This novel perspective provides an avenue for both finer-grained diagnostics of large language models and more deliberate design of mid-layer representations for downstream tasks.

# 3. A Unified Framework for Neural Representations

> **Key Takeaway:** Matrix-based entropy unifies seemingly disparate metrics of representation quality, providing a single theoretical lens for analyzing compression, geometry, and invariance.

A central challenge in analyzing internal representations is determining *how* to assess their quality. Although existing work draws on numerous ideas—from mutual information to geometric manifold analysis to invariance under augmentations—these threads can seem disparate. In this section,

we consolidate them into a *unified theoretical framework* that shows *how* these seemingly different metrics connect and *why* they collectively measure "representation quality."

## 3.1. Notation and Motivation

Consider a neural network that maps inputs $\mathbf{x}$ (e.g., tokens in a sequence) to internal hidden states $\mathbf{Z}$. We denote $\mathbf{Z} \in \mathbb{R}^{N \times D}$ as a matrix of $N$ data samples (or tokens) in $D$ dimensions. Some key questions arise:

1. *How compressed* are these representations?

2. *How robust* are they to perturbations or augmentations?

3. *How do they geometrically organize* different inputs?

Answers to these questions can illuminate which layers strike the right balance between preserving relevant features and discarding noise.

## 3.2. Matrix-Based Entropy: A Common Theoretical Thread

We focus on a key quantity known as *matrix-based entropy* (Giraldo et al., 2014; Skean et al., 2023), which applies directly to the Gram matrix $\mathbf{K} = \mathbf{Z}\mathbf{Z}^\top$. Let $\{\lambda_i(\mathbf{K})\}$ be the (nonnegative) eigenvalues of $\mathbf{K}$. For any order $\alpha > 0$, define:

$$S_\alpha(\mathbf{Z}) \;=\; \frac{1}{1-\alpha}\, \log\!\left( \sum_{i=1}^{r}\!\left(\tfrac{\lambda_i(\mathbf{K})}{\mathrm{tr}(\mathbf{K})}\right)^{\!\alpha} \right), \qquad (1)$$

where $r = \mathrm{rank}(\mathbf{K}) \le \min(N, D)$. Intuitively, if only a few eigenvalues dominate, $S_\alpha(\mathbf{Z})$ is *small*—indicating a highly compressed representation. Conversely, if $\mathbf{Z}$ is spread out across many principal directions, $S_\alpha(\mathbf{Z})$ is *large*. By varying $\alpha$, one smoothly transitions between notions like collision entropy ($\alpha = 2$) and von Neumann entropy ($\alpha \to 1$). We will typically use $\alpha = 1$ for simplicity.

**Bridging geometry, invariance, and feature locality.** A key benefit of matrix-based entropy is that it unifies multiple representational perspectives:

- **Compression** or *information content:* A handful of large eigenvalues in $\mathbf{K} = \mathbf{Z}\mathbf{Z}^\top$ indicates that $\mathbf{Z}$ is low-rank, i.e. the model has collapsed much of the input variation into fewer dimensions. In contrast, a more uniform eigenvalue spectrum implies higher-entropy, more diverse features.

- **Geometric smoothness:** If tokens within a prompt follow a trajectory in embedding space with *sharp turns*, that curvature can manifest as skewed eigenvalue spectra (Hosseini & Fedorenko, 2023). Curvature also differentiates *local* transitions (token-to-token) from *global* structural patterns across longer segments or entire prompts.

- **Invariance under augmentations:** Metrics like InfoNCE (Oord et al., 2018) and LiDAR (Thilak et al., 2024) effectively measure whether augmentations of the same sample (e.g. character swaps) map to *similar* embeddings. Strong invariance corresponds to stable clustering in $\mathbf{Z}\mathbf{Z}^\top$, which again depends on the distribution of eigenvalues and how local vs. global features are retained or discarded.

Thus, evaluating $S_\alpha(\mathbf{Z})$ provides a single lens for assessing "representation quality" across compression, geometric structure, and invariance—and highlights how *both* local details and global patterns are organized.

## 3.3. Representation Evaluation Metrics

> **Key Takeaway:** Information-theoretic, geometric, and invariance-based metrics offer complementary perspectives on representation quality that can all be understood through matrix-based entropy.

We now introduce the seven representation evaluation metrics used in our experiments, grouped into three broad categories: (1) *information-theoretic*, (2) *geometric*, and (3) *augmentation-invariance*. All relate back to the Gram matrix $\mathbf{K}$ and hence to Eq. (1).

### 3.3.1. INFORMATION-THEORETIC METRICS

**Prompt Entropy.** Following Wei et al. (2024), we apply matrix-based entropy (Eq. 1) to the token embeddings *within a single prompt*. This *prompt entropy* quantifies how widely tokens are spread in the embedding space. Higher entropy indicates more diverse, less redundant token-level features; lower entropy implies stronger compression.

**Dataset Entropy.** We can also aggregate embeddings *across $N$ prompts* by taking the mean token embedding of each prompt to form $\overline{\mathbf{Z}} \in \mathbb{R}^{N \times D}$. Applying entropy to $\overline{\mathbf{Z}}$ yields a *dataset*-level measure of global diversity—revealing how distinctly the model separates different inputs.

**Effective Rank.** (Roy & Vetterli, 2007) can be shown to be a lower bound to $\exp(S_1(\mathbf{Z}))$, highlighting how dimensionality effectively shrinks if the representation is strongly compressed. We prove this connection in Theorem 1. This has implications for popular representation evaluation metrics such as RankMe (Garrido et al., 2023) and LiDAR (Thilak et al., 2024), which are both inspired by Effective Rank.

### 3.3.2. GEOMETRIC METRICS

**Curvature.** Proposed by Hosseini & Fedorenko (2023), *curvature* captures how sharply the token embeddings turn when viewed as a sequence in $\mathbb{R}^D$. For a prompt of length $L$,

let $\mathbf{v}_k = \mathbf{z}_{k+1} - \mathbf{z}_k$ be the difference between consecutive tokens. The average curvature is:

$$\bar{C} = \frac{1}{L-2} \sum_{k=1}^{L-2} \arccos\left( \frac{\mathbf{v}_{k+1}^\top \mathbf{v}_k}{\|\mathbf{v}_{k+1}\|\|\mathbf{v}_k\|} \right).$$

Higher curvature means consecutive tokens shift direction abruptly and more local level features; lower curvature suggests a smoother trajectory and global level features.

### 3.3.3. AUGMENTATION INVARIANCE METRICS

Lastly, we assess how stable the model's representations are to small perturbations of the same input (e.g., random character swaps, keyboard-level changes; see Appendix). Suppose a prompt $p_i$ is augmented into $p_i^{(a)}$ and $p_i^{(b)}$. After embedding these, we compare the row vectors in $\mathbf{Z}_1, \mathbf{Z}_2 \in \mathbb{R}^{N \times D}$ under different scoring criteria:

**InfoNCE.** This self-supervised objective (Oord et al., 2018) encourages matched samples to lie close in embedding space while pushing unmatched samples away. A *lower* InfoNCE loss indicates stronger invariance to augmentation.

**LiDAR.** LiDAR (Thilak et al., 2024) uses a linear discriminant approach that measures within-class versus between-class scatter. Treating each prompt as its own class, LiDAR checks how well augmentations form tight clusters.

**DiME.** Similarly, DiME (Skean et al., 2023) is grounded in matrix-based entropy. It compares real paired samples against random pairings to estimate how uniquely aligned correct augmentations are.

## 3.4. Core Theoretical Results

> **Key Takeaway:** Our theoretical framework establishes concrete connections between representation entropy and downstream performance through properties like effective rank and invariance.

Here, we summarize key statements that justify why these metrics meaningfully measure representation quality. We refer to the appendix G for details and proofs. Beyond serving as a unifying view, matrix-based entropy also connects to foundational concepts like majorization, Schur concavity, and mutual information. Furthermore, we can directly relate the eigenvalue entropy to the matrix entropy, most naturally via the Effective Rank (Roy & Vetterli, 2007). The following theorem makes this connection explicit.

**Theorem 1** (Lower Bound via Effective Rank)**.** *For Shannon-based entropy ($\alpha \to 1$),*

$$\mathrm{EffRank}(\mathbf{Z}) \leq \exp\big(S_1(\mathbf{Z})\big),$$

*meaning a large effective rank implies a high entropy.*

Under appropriate conditions on the data distribution and model, we can show connections between prompt entropy and dataset entropy via the following scaling behaviors:

**Theorem 2** (Informal)**.**

1. *If* prompt entropy *remains near its maximum for* all *prompts, then the* dataset entropy $S_2(\overline{\mathbf{Z}}\,\overline{\mathbf{Z}}^\top)$ *grows on the order of* $\log\left(\frac{L^2}{N}\right)$.

2. *If* prompt entropy *instead stays near its minimum for* all *prompts, then dataset entropy grows more slowly, on the order of* $\log\left(\frac{L^2}{N^3}\right)$.

In short, high token-level (prompt) diversity encourages broader *global* diversity in the dataset-level embeddings, whereas over-compressing token representations can limit how effectively different prompts separate. Our subsequent analysis connects these ideas to self-supervised objectives like InfoNCE, which also tie higher entropy to stronger robustness and discriminability in the learned representations.

**Theorem 3** (Dataset Entropy Bounds InfoNCE)**.** *For data* $X$ *and representation* $Z(X)$, *the InfoNCE loss on* $N$ *samples satisfies:*

$$\log(N) - \text{InfoNCE} \;\leq\; I(X; Z) \;\leq\; H(Z),$$

*where* $H(Z)$ *is interpretable as matrix-based entropy at the dataset level. Hence, reducing InfoNCE implies learning a higher-entropy (and thus often more robust) representation.*

**Practical outlook.** Overall, our theoretical analysis shows that *compression* (entropy), *geometry* (curvature, rank), and *invariance* (e.g. InfoNCE) are all facets of how the Gram matrix $\mathbf{ZZ}^\top$ distributes variance. Examining these metrics across different layers reveals exactly where a network "prunes" redundancy (low entropy) versus preserving essential distinctions (high entropy). This unified perspective also facilitates cross-architecture comparisons (e.g. transformers vs. SSMs) by highlighting how each architecture organizes information internally. Beyond offering a theoretical foundation, it provides a practical blueprint for diagnosing, tuning, and improving hidden-layer representations.

## 4. Empirical Results

In this section, we empirically test our theoretical framework through extensive experiments across architectures, scales, and training regimes. We focus on three key questions:

- **Do intermediate layers consistently outperform final layers across diverse downstream tasks?**

- **How do these intermediate representations differ across architectures, training stages, and scales?**

- **How does post-training methods (e.g., fine-tuning and chain-of-thought) reshape representations?**

### 4.1. Downstream Task Performance

> **Key Takeaway:** Intermediate layers of language models consistently outperform final layers across all architectures and tasks, challenging the conventional wisdom of using final-layer representations.

In this section, we use intermediate layers for downstream embedding tasks and employ our unified framework from Section 3, measuring all the embeddings across all layers.

#### 4.1.1. EXPERIMENTAL SETUP

**Models** We evaluate three distinct architectural families: Pythia and Llama3 (decoder-only transformers) (Biderman et al., 2023; Dubey et al., 2024), Mamba (state space model) (Gu & Dao, 2024), BERT (encoder-only transformer) (Devlin et al., 2019) and LLM2Vec models (bidirectional attention) (Behnam Ghader et al., 2024).

**Tasks** We test each layer's embeddings on 32 tasks from the Massive Text Embedding Benchmark (MTEB) (Muennighoff et al., 2022), spanning classification, clustering, and reranking dor a comprehensive evaluation across various tasks. We refer to the Appendix for details.

#### 4.1.2. INTERMEDIATE LAYERS OFTEN OUTPERFORM FINAL LAYERS

Are final-layer embeddings indeed optimal for downstream tasks? In Figure 1, we compare average performance on MTEB tasks across all layers of the three models.

**Key observation.** *In nearly every task, some intermediate layer outperforms the final layer.* The absolute improvement ranges from 2% to as high as 16% on average, and the best layer often resides around the mid-depth of the network. This phenomena is consistent across all the different architectures. This confirms emerging observations in recent work for generation tasks (Bordes et al., 2023; El-Nouby et al., 2024; Chen et al., 2020; Fan et al., 2024) and extends them to a wider range of benchmarks and tasks.

**Why do these layers matter?** From our theoretical perspective, intermediate layers appear to strike a balance between retaining sufficient information (avoiding over-compression) and discarding low-level noise. Later in Section 4.2, we show that these sweet spots are not random but tied to how intermediate layers are processing information.

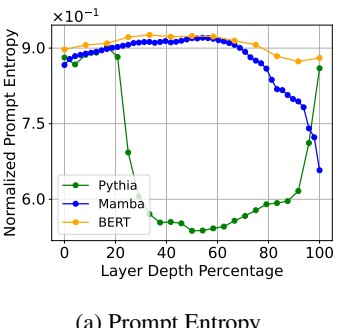
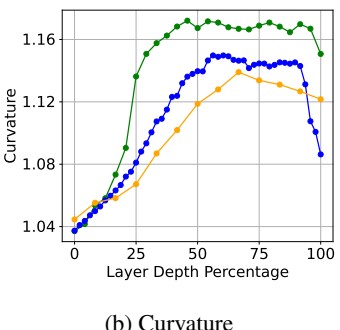
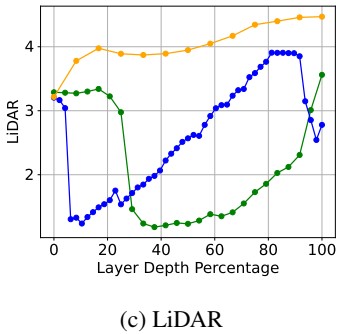

(a) Prompt Entropy   (b) Curvature   (c) LiDAR

Figure 2: **Pythia and Mamba's intermediate layers show pronounced changes in representation quality metrics, while BERT's remain more stable.** Three representation evaluation metrics calculated on the wikitext dataset for every layer in Pythia-410M, Mamba 370M, and BERT-base architectures. The x-axis denotes layer depth as a percentage, allowing fair comparison between models with different layer counts.

### 4.1.3. LAYER-WISE METRICS CORRELATE WITH DOWNSTREAM PERFORMANCE

To validate our framework, we analyze how each evaluation metric correlates with downstream performance. Figures 3 and 8 show distance correlations between metrics and task scores for Pythia-410M. We find that all metrics exhibit strong relationships with downstream performance. Among them, curvature, DiME, and InfoNCE stand out with particularly high correlations. These associations remain robust across different correlation measures, including Spearman and Kendall, reinforcing the reliability of our findings.

Our results suggest that our metrics capture some aspects of intermediate representations that contribute to downstream utility. In Appendix E, we leverage these strong correlations to select high-performing layers in an unsupervised manner, following (Agrawal et al., 2022; Garrido et al., 2023; Thilak et al., 2024). In short, we can identify an intermediate layer that surpasses the final layer in downstream performance—without using any task-specific labels. For instance, using DiME-based layer selection leads to a 3% average improvement in MTEB scores for the Pythia-410M model.

### 4.2. Architectural and Scale Differences

> **Key Takeaway:** Different architectures exhibit distinct patterns of information compression. Autoregressive models show mid-layer bottlenecks while bidirectional models maintain more uniform trends.

Aside from strong correlations with downstream performance, we can use our evaluation framework to assess the internal behaviors of LLMs. In both this section and Section 4.3, we use WikiText-103 (Merity et al., 2017) for analyzing our representation metrics on standard textual data. To investigate how architecture and model size influ-

ence representation quality, we compare three fundamentally different LLM variants—BERT (encoder-only), Pythia (decoder-only), and Mamba (state-space model)—and then scale up Pythia to observe emerging trends.

**Encoder vs. Decoder vs. SSM.** Figure 2 shows how prompt entropy, curvature, and augmentation metrics evolve across each model's layers. BERT, which encodes the entire input bidirectionally, generally maintains high entropy across layers, suggesting minimal compression: the model can see all tokens at once and need not discard as much information. By contrast, the decoder-only Pythia exhibits a strong mid-layer entropy dip, reflecting its autoregressive objective's tendency to filter or prune non-local details in the middle of the network. As a result, Pythia's "sweet spot" for downstream tasks often lies around mid-depth, where it

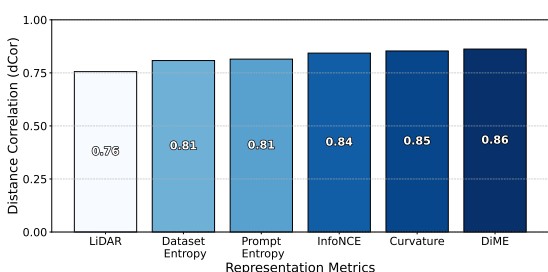

Figure 3: **Relationship between representation metrics and task performance averaged across layers for Pythia 410M.** Using distance correlation (dCor), we see strong associative relationships across the board with DiME exhibiting the strongest relationship with downstream performance. We use dCor due to its robustness and ability to measure both linear and non-linear relationships (dCor $\in [0, 1]$ with 0 indicating statistical independence and 1 indicating strong dependency). We defer additional results to the Appendix.

balances essential context and compression. Mamba, meanwhile, processes sequences through a state-space approach that yields flatter, more uniform curves across depth: it neither retains as much information as BERT nor compresses as aggressively as Pythia's mid-layers. These conclusions align with Razzhigaev et al. (2024) which showed a flat layer-wise anisotropy for encoder models and a spike in intermediate layer anisotropy for decoder models.

**Scaling Size Effects.** In Figure 12, we analyze Pythia models ranging from 14M to 1B parameters. Larger models display more pronounced intermediate compression (entropy dips), indicating a heightened ability to distill relevant features. We also observe smoother token trajectories (lower curvature) and stronger invariance (higher LiDAR), consistent with findings that bigger models more effectively filter noise and capture long-range dependencies. These trends reinforce why performance peaks in the middle of the network: larger models hold more capacity to compress intermediate representations, yet still preserve crucial semantic details.

**Finetuning Effects** In Figure 13, we study how finetuning affects the internal representations of Llama3 (Dubey et al., 2024). We compare the baseline Llama3-8B to two finetuned LLM2Vec models (Behnam Ghader et al., 2024). The LLM2Vec-mntp-unsup-simcse model enables bidirectional attention in Llama3 and goes through two unsupervised training phases to improve Llama3's performance on embedding tasks. The LLM2Vec-mntp-supervised adds an additional supervised finetuning phase. It is clear that both finetuned models have improved augmentation invariance. Furthermore, the unsupervised model has higher prompt entropy than Llama3 while the supervised model has less.

**Layer-Level Analysis of Transformer Sub-Components.** While our experiments treat each transformer layer as a single unit, transformer blocks are composed of multiple sub-layers (pre-attention normalization, self-attention, residuals, MLPs). By measuring entropy after each sub-layer, we find in Figure 15 that *residual connections* drive the mid-network compression observed in Section 4.2. Specifically:

- **Sub-layers *before* residuals** (e.g. pre-attention, attention scores, or MLP pre-residual outputs) often show only mild compression.

- **Residual sub-layers** exhibit a pronounced drop in entropy, indicating a significant filtering of information. A concurrent study (Csordás et al., 2025) observed a decrease in the residual stream norm in the second half of decoder models, reinforcing our findings.

The strong entropy "valley" at intermediate layers is tied to how residual paths merge new signals with the existing hidden state. This aligns with prior work indicating that residuals act as a regularizer (Marion et al., 2024), smoothing out spurious components in hidden representations.

### 4.3. Impact of Training Progression

> **Takeaway:** Significant changes during training occur in intermediate layers and early layers stabilize quickly, supporting the detokenization hypothesis.

We measure Pythia's metrics at multiple checkpoints to understand how layer-wise representations evolve throughout training (Figures 4 and 11). Two main observations emerge:

**Intermediate Layers Undergo the Most Change.** The largest shifts in representation quality occur in mid-depth layers. Specifically, *prompt entropy* steadily decreases there as training progresses, implying that intermediate layers increasingly compress and abstract the input. Meanwhile, *LiDAR* scores are minimal in these same layers. Likewise, *curvature* becomes smoother in the middle of the network, suggesting the model refines its internal structure to capture longer-range or more nuanced patterns in language.

**Early Layers Stabilize Quickly.** In contrast to intermediate layers, the earliest layers change very little after the initial phase of training. This observation aligns with the "detokenization" hypothesis of Lad et al. (2024), which posits that the main functional role of early layers is to convert raw tokens into a basic embedding space. This idea is closely related to the "shared task" layers of Zhao et al. (2024), introduced in the context of instruction tuning on diverse tasks. In particular, they show that the first nine layers of LlaMA 2 7B (Touvron et al., 2023) perform general task-agnostic operations. As a result, the most substantial changes to representations, such as enhanced compression, are driven primarily by the intermediate layers, reinforcing their importance for learning robust, high-level features.

### 4.4. Impact of Chain-of-Thought Finetuning

> **Key Takeaway:** CoT finetuning enables models to maintain richer context throughout their layers.

Recent work has highlighted Chain-of-Thought (CoT) finetuning as a powerful strategy for improving reasoning capabilities (Arefin et al., 2025; DeepSeek-AI, 2025). To examine its effects on representations, in Figure 5 we compare Qwen 2.5 and Qwen 2.5-Math (Yang et al., 2024), where the latter underwent additional math pretraining and CoT finetuning. Measuring token-level prompt entropy across sequence length reveals that the finetuned model maintains higher entropy with lower variance across examples.

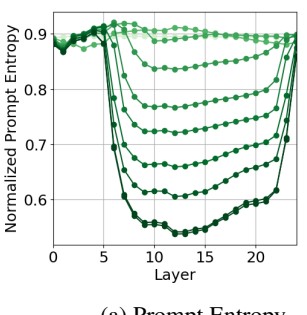
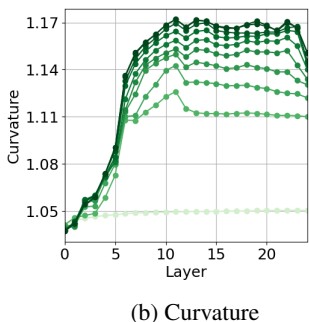
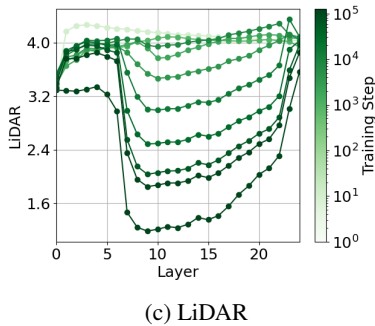

(a) Prompt Entropy        (b) Curvature        (c) LiDAR

Figure 4: **Strong trends in intermediate behavior emerge during training** Representation evaluation metrics across layers at various Pythia-410M training checkpoints, ranging from step 1 to the final step at 143k. The x-axis is the model layer, showing how training affects different layers, while the colors are different checkpoints during training.

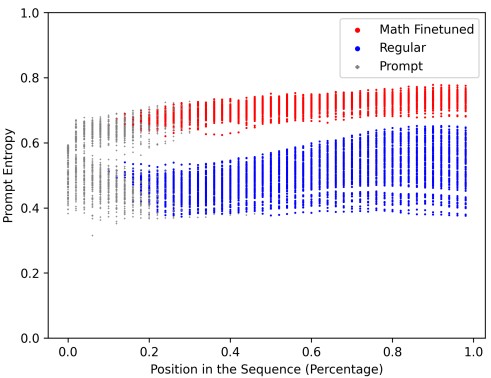

Figure 5: Token-level prompt entropy across sequence lengths for Qwen 2.5 and Qwen 2.5-Math. The base model (Qwen 2.5) exhibits greater prompt compression, while the finetuned (Qwen 2.5-Math) has higher entropy, indicating more information retention.

These findings suggest that CoT finetuning encourages models to preserve more context throughout their hidden layers, enabling better multi-step reasoning. Our framework provides a quantitative lens into how CoT fine-tuning pushes models to maintain richer internal representations across sequences, explaining its effectiveness in multi-step tasks. While CoT traces can be inspected directly in these models, our approach is particularly valuable for analyzing models that reason in continuous latent space (Hao et al., 2024).

## 5. Extreme Input Conditions

To better probe the underlying factors affecting representation quality, we inspect each layer's responsiveness to different input types. We use Pythia-410M on three types of *extreme* prompts and measure prompt entropy across layers (Figure 6). We prove examples of these prompts in Appendix F. Overall, we find that:

1. **Token repetition compresses intermediate layers.** As $p$ increases (i.e., more repeated tokens), *prompt entropy* decreases sharply in mid-depth layers, suggesting that the model recognizes/encodes repetitive patterns and discards redundancy in its internal representation.

2. **Random tokens inflate early-layer entropy.** Adding token-level randomness, increases entropy significantly in early layers, revealing their sensitivity to noise. In contrast, deeper layers are more robust.

Overall, these results confirm that *intermediate layers* play a major role in handling complex or unusual inputs, selectively compressing or filtering out repetitive patterns while retaining crucial distinctions. Early layers are more sensitive to noise and the incremental benefit of adding more tokens diminishes with prompt length. This behavior highlights the diverse ways in which different layers balance the trade-off between preserving and discarding information, underscoring the significance of intermediate representations.

## 6. Comparison to Vision Transformers

Do our findings extend to other domains like computer vision? Vision models employ diverse architectures and training objectives from fully supervised learning to self-supervised methods, and from bidirectional to autoregressive encoders. Their diversity provides an ideal testbed to examine how well our findings generalize and how different training objectives shape internal representations.

We examine several representative vision approaches: **ViT** (Dosovitskiy et al., 2021), a *supervised* transformer trained on labeled data; **CLIP** (Radford et al., 2021), a *weakly supervised* image encoder; **BEiT** (Bao et al., 2022), a *self-supervised* encoder that reconstructs masked patches; **DINOv2** (Oquab et al., 2024), a self-supervised approach leveraging augmentations and exponential moving average teachers; **MAE** (He et al., 2022), a self-supervised approach

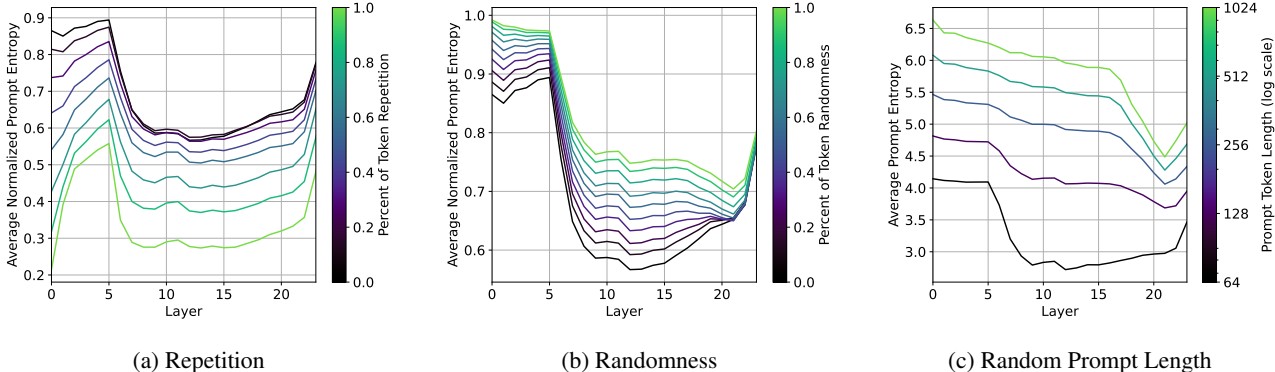

(a) Repetition                    (b) Randomness                    (c) Random Prompt Length

Figure 6: **Prompt entropy across layers of Pythia 410M under various extreme input conditions.** (a) Increasing token repetition leads to decreased entropy in intermediate layers. (b) Increasing token randomness results in higher entropy, especially in initial layers. (c) Unnormalized prompt entropy increases with prompt length due to the larger number of tokens. These results demonstrate how the model's internal representations adapt to different types of input perturbations.

that reconstructs images from masked patches; **AIM** (El-Nouby et al., 2024), an *autoregressive* transformer that predicts the next patch in an image sequence (GPT-style next-token prediction); and **AIMv2** (Fini et al., 2025), which extends AIM with a multimodal next-token prediction task. In Figure 14, we evaluate every model layer on ImageNet-1k with attention probing and our suite of metrics.

**AIM exhibits behavior similar to language models.** AIM, which predicts image patches sequentially, exhibits the same entropy "valley" and accuracy peak at intermediate layers that we observed in language models like Pythia. This pattern suggests that autoregressive training, whether over text tokens or image patches, consistently creates a mid-depth information bottleneck. The sequential prediction constraint forces models to compress non-local contextual information early in processing, then selectively re-expand the most relevant features for accurate prediction. AIM's strong intermediate performance was first noted in (El-Nouby et al., 2024). Interestingly, while the AIMv2 model does not show improved intermediate accuracy, it still produces an entropy valley. We hypothesize this difference is due to the multimodal text-vision pretext task, which may alter information compression dynamics.

**Vision transformers behave differently from language models.** All models except for AIM exhibit strictly increasing downstream accuracy toward final layers. Similar trends have been shown for ResNets (Sorscher et al., 2022), where few-shot classification error is strictly decreasing across layers. Most non-autoregressive vision models show steadily *increasing* dataset entropy. The notable exception is BEIT, which exhibits a substantial intermediate dip. Taken together, the results suggest that without an autoregressive objective, vision transformers have less need for drastic

transformations at mid-depth.

**Autoregression as the driving factor.** The strong mid-layer compression observed in LLMs seems to be not purely a property of "sequential token data" vs. "image patch data," but rather a byproduct of pretraining. While various self-supervised (or fully supervised) objectives in vision foster more uniform feature building across layers, autoregressive vision models develop similar mid-layer bottlenecks that we see in language. Thus, the objective design–whether or not a model is autoregressive—appears crucial in shaping layer-wise representation quality, regardless of domain.

# 7. Discussion and Conclusion

We investigated the representation quality of intermediate layers in LLMs and their role in downstream task performance. We introduced a unified framework of evaluation metrics, establish theoretical connections among them, and apply these metrics to analyze transformer-based architectures, SSMs, and vision models. A key phenomenon unveiled by prompt entropy was an information bottleneck in the middle layers of autoregressive transformers in both vision and language domains. Furthermore, we show that intermediate layers often surpass final layers in representation quality, holding implications for feature relevance and extraction. DiME, curvature, and infoNCE correlate well with downstream performance, suggesting a fundamental connection between representation and generalizability.

In conclusion, our work studies the internal representation dynamics in LLMs, offering theoretical and empirical insights as well as practical implications for optimizing model design and training strategies. Future work should further investigate the underlying causes of intermediate layer compression and do explicit finetuning to control compression.

## Impact Statement

Our paper studies the inner workings of large language models with findings that may challenge typical assumptions about the importance of intermediate layers in large language models and the representations they learn. Our findings suggest that representations from these layers can yield better performance on a variety of downstream tasks, which can have implications for model interpretability, robustness, and efficiency.

From an ethical standpoint, the ability to leverage intermediate-layer representations could impact fairness and bias considerations in evaluating model performance or in model deployment. By helping better identify latent features and representations, our approach may amplify latent biases. We welcome and encourage future work to explore methods that can ensure that intermediate-layer representations do not disproportionately reinforce biases or lead to unintended disparities in real-world applications.

## Acknowledgements

We thank the anonymous reviewers for their valuable feedback, which helped improve the clarity and presentation of our results. We are also grateful to Diego Doimo, Artemii Novoselov, Jhoan Keider Hoyos Osorio, Luis Sanchez, and Matteo Saponati (listed alphabetically) for fruitful discussions and helpful pointers to related literature. Oscar Skean is supported by the Office of the Under Secretary of Defense for Research and Engineering under award number FA9550-21-1-0227.

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

# A. Architectural Details

In this section, we elaborate on the specific architectures of transformers and State Space Models (SSMs). We outline the mathematical foundations, including the weight matrices, attention mechanisms for transformers, and the state transition matrices for SSMs. Detailed equations and parameter configurations are provided to facilitate replication and deeper understanding.

## A.1. Transformer

The transformer architecture (Vaswani et al., 2017) utilizes self-attention mechanisms. Given an input $\mathbf{x}$, the key ($\mathbf{K}$), query ($\mathbf{Q}$), and value ($\mathbf{V}$) matrices are computed as:

$$\mathbf{Q} = \mathbf{x}\mathbf{W}_Q, \quad \mathbf{K} = \mathbf{x}\mathbf{W}_K, \quad \mathbf{V} = \mathbf{x}\mathbf{W}_V, \quad (2)$$

where $\mathbf{W}_Q, \mathbf{W}_K \in \mathbb{R}^{d \times d_k}$ and $\mathbf{W}_V \in \mathbb{R}^{d \times d_v}$ are learned weights.

The attention weights are calculated using:

$$\mathbf{A} = \text{softmax}\left(\frac{\mathbf{Q}\mathbf{K}^\top}{\sqrt{d_k}} + \mathbf{M}\right), \quad (3)$$

where $\mathbf{M}$ is a mask to enforce causality in autoregressive tasks.

The output is then:

$$\mathbf{y} = \mathbf{A}\mathbf{V}. \quad (4)$$

## A.2. State Space Models

SSMs (Gu & Dao, 2024) model sequences using recurrent dynamics. The hidden state $\mathbf{h}_t$ and output $\mathbf{y}_t$ at time $t$ are updated as:

$$\mathbf{h}_t = \mathbf{A}\mathbf{h}_{t-1} + \mathbf{B}\mathbf{x}_t, \quad (5)$$
$$\mathbf{y}_t = \mathbf{C}\mathbf{h}_t + \mathbf{D}\mathbf{x}_t, \quad (6)$$

where $\mathbf{A} \in \mathbb{R}^{n \times n}$, $\mathbf{B} \in \mathbb{R}^{n \times d}$, $\mathbf{C} \in \mathbb{R}^{d \times n}$, and $\mathbf{D} \in \mathbb{R}^{d \times d}$ are learned parameters.

# B. Discussion on Prompt Entropy

The first measure of token embedding diversity we call prompt entropy. This entropy is measured on the intermediate tokens and captures how diverse the token representations are.

We follow the work of (Wei et al., 2024) and use $\alpha$-order matrix-based entropy (Giraldo et al., 2014; Skean et al.,

2023; 2024), which serves as a tractable surrogate for traditional Rényi's $\alpha$-order entropy (Rényi, 1961). The quantity is calculated using a similarity kernel $\kappa$ on a batch of samples drawn from a distribution, without making explicit assumptions on what the true distribution is. The choice of kernel $\kappa$ is flexible and can be any infinitely divisible kernel such as the Gaussian kernel, linear kernel, or Laplacian kernel, among others. For this work, we restrict ourselves to the linear kernel $\kappa(a, b) = ab^T$. This choice is motivated by the linear representation hypothesis (Park et al., 2024b) which finds that large language model representations encode high-level concepts such as truth (Burns et al., 2023), honesty (Mallen & Belrose, 2024), and part-of-speech (Mamou et al., 2020) in linearly separable manifolds.

The equation for matrix-based entropy was previously defined in Eq. 1. One way to interpret Eq. 1 is as the $\alpha$-order Rényi entropy of the Gram matrix eigenvalues[2]. Notice how each eigenvalue is divided by $\text{tr}(\mathbf{K_Z})$ before being raised to the $\alpha$ power. This is so that the eigenvalues of $\mathbf{K_Z}$ sum to one (because $\text{tr}(\cdot) = \sum_{i=1}^n \lambda_i(\cdot)$), which is a necessary condition to treat the eigenvalues as a probability distribution. Futhermore, each eigenvalue of $\mathbf{K_Z}$ signifies the variance of samples in a particular principal component direction (Scholkopf & Smola, 2018). If entropy is low, then

---

[2]The non-zero eigenvalues of the Gram matrix $ZZ^T$ are equivalent to those of the covariance matrix $Z^T Z$. Using the covariance matrix instead of the Gram matrix in Eq. 1 makes no difference and is more computationally efficient if $D < N$.

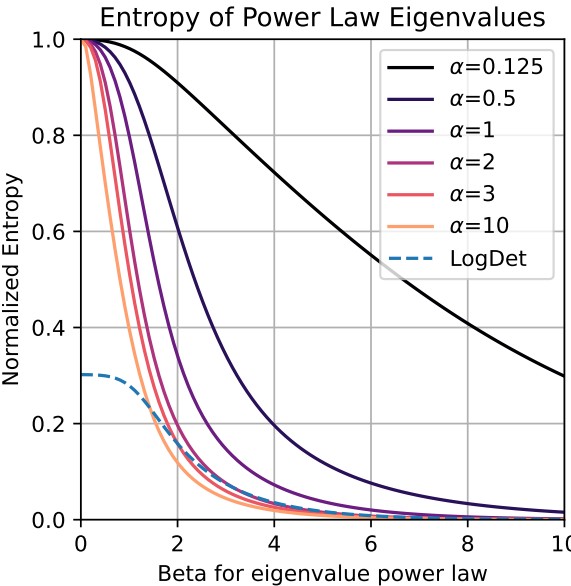

Figure 7: The behavior of Eq. 1 for varying values of $\alpha$ on Gram matrices with eigenvalues distributed with a $\beta$-power law such that $\lambda_i = i^{-\beta}$.

the eigenvalues form a heavy-tail distribution which implies that a few components dominate the variance of samples in $Z$. On the other hand, at maximum entropy, the eigenvalues form a uniform distribution and samples are spread equally in all directions. Matrix-based entropy is reminiscent of the LogDet entropy which uses the determinant of $\mathbf{K_Z}$ to capture how much "volume" a dataset occupies (Shwartz-Ziv et al., 2023; Zhouyin & Liu, 2021). The LogDet entropy is given by $S_{\text{LogDet}}(Z) = \log \det(\mathbf{K_Z}) - \log 2$. One can use Jensen's inequality to show that the LogDet entropy is a lower bound of Eq 1 when $\lim_{\alpha \to 1}$ (Appendix J.4 of (Shwartz-Ziv et al., 2023)).

Depending on the choice of $\alpha$, several special cases of matrix-based entropy can be recovered. In particular, when $\lim_{\alpha \to 1}$ it equals Shannon entropy (also referred to as von Neumann entropy in quantum information theory (Bach, 2022; Boes et al., 2019)), and when $\alpha = 2$ it equals collision entropy. Interestingly, the case of $\alpha = 2$ can be calculated without explicit eigendecomposition (Skean et al., 2024). We show in the Appendix Figure 7 how varying values of $\alpha$ affect the matrix-based entropy of Gram matrices with eigenvalues distributed with a $\beta$-power law such that $\lambda_i = i^{-\beta}$. It is shown that for larger values of $\alpha$, smaller eigenvalues contribute more to the entropy.

## C. Dataset Details

### C.1. Wikitext Dataset

We used the wikitext dataset (Merity et al., 2017) for the majority of our experiments in Sections 4.2 and 5. This was downloaded from **Salesforce/wikitext** on huggingface. The dataset consists of 100 million tokens scraped from the Featured articles on wikipedia. We filtered out prompts which were less than 30 tokens or were wikipedia section headings.

### C.2. MTEB

The 32 tasks we used from the Massive Text Embedding Benchmark (MTEB) are detailed in Table 1. They are English language tasks covering clustering, classification, reranking, and sentence-to-sentence.

## D. Prompt Augmentations

For the augmentation-invariance metrics such as infoNCE, LiDAR, and DiME, we use the NLPAug library (Ma, 2019) to augment our prompts. We use three types of augmentations.

- The SplitAug augmentation randomly splits words into two parts by adding a space.

- The RandomCharAug augmentation randomly inserts, substitutes, swaps, or deletes characters.

- The Keyboard augmentation randomly substitutes characters with other characters that are at a distance of one as measured on a QWERTY keyboard. For instance, the character "k" may be replaced with "i", "l", "m", or "j".

We use the pseudocode below to do our augmentations using three types of augmentations, using the default library settings for each type. When computing augmentation-invariance metrics like infoNCE or DiME, we use the two augmented prompts rather than using one augmented prompt alongside the original prompt. Note that these augmentations may change the token length $T$ of a prompt.

```
aug = naf.Sequential([
    naw.SplitAug(p=0.3),
    nac.RandomCharAug(p=0.3),
    nac.KeyboardAug(p=0.3),
])
(aug_A, aug_B) = aug.augment(prompt,
    num_augmentations=2)

prompt -> "The quick brown fox jumps
    over the lazy dog."

aug_A -> "The quDUk b rown fox wEmps o
    ver the l azy dog."
aug_B -> "The qTuXi bro wn fox uVm)s
    ob3r the la_k dog."
```

## E. Using Evaluation Metrics as a Performance Proxy

We previously demonstrated strong correlations between our unsupervised evaluation metrics and downstream performance. These correlations can be exploited to select high-performing layers for a given task entirely without supervision, as suggested by prior work (Agrawal et al., 2022; Garrido et al., 2023; Thilak et al., 2024).

In Figure 2, we apply this unsupervised layer selection approach to Pythia-410M and LLM2Vec-8B using the 32-task MTEB benchmark introduced in Section 4.1. Rather than computing task accuracies for every layer, we compute DiME, infoNCE, and dataset entropy for each task across all layers in a single forward pass. For each task, we then select the layer that minimizes one of these metrics—leveraging their negative correlation with downstream performance.

This straightforward yet effective method yields substantial performance improvements with no supervision. For example, DiME-based layer selection boosts the average MTEB score of Pythia-410M by 3%.

# F. Extreme Prompts

## F.1. Increasing Repetition

We take regular prompts from the wikitext dataset, tokenize them, and then for each token we randomly replace it with probability $p$. We draw replacements tokens by sampling a random token from within the prompt. We show examples below for varying levels of $p$.

- ($p = 0$)   Mint records indicate the first gold dollars were produced on May 7...

- ($p = 0.1$) Mint records indicate the first gold dollars were Mint Mint May 7...

- ($p = 0.5$) Mint records Mint Mint Mint gold dollars were Mint Mint Mint 7...

- ($p = 1.0$) Mint Mint Mint Mint Mint Mint Mint Mint Mint Mint Mint Mint Mint...

## F.2. Increasing Randomness

We take regular prompts from the wikitext dataset, tokenize them, and then for each token we randomly replace it with probability $p$. We draw replacements uniformly from the tokenizer distribution. We show examples below for varying levels of $p$. Unlike the character-level random noise added to prompts in Section with random noise discussed in Appendix D which might change the number of tokens $T$ of the prompt, the token-level random noise used here does not do so.

- ($p = 0$)   Mint records indicate the first gold dollars were produced on May 7...

- ($p = 0.1$) Mint records indicate salivary first gold dollars were produced on May NaCl...

- ($p = 0.5$) Mint records Dallas actively first dollars persufors on Mayder129 18...

- ($p = 1.0$) arf emulsion minorensteinorianmega_TOStack potsRecip Installifykeeping...

# G. Theorems

**Definition 1.** *(Majorization) Let $p, q \in R^n$ be nonnegative vectors such that $\sum_{i=1}^{N} p_i = \sum_{i=1}^{N} q_i$. We say that $q$ majorizes $p$, denoted by $p \preccurlyeq q$, if their ordered sequences $p_{[1]} \geq \cdots \geq p_{[n]}$ and $q_{[1]} \geq \cdots \geq q_{[n]}$ satisfy:*

$$\sum_{i=1}^{k} p_{[i]} \leq \sum_{i=1}^{k} q_{[i]} \quad for \quad k = 1, \cdots, n \qquad (7)$$

**Definition 2.** *(Schur-Convexity) A real-valued function $f$ on $\mathbb{R}^n$ is called Schur-convex if $p \preccurlyeq q \implies f(p) \leq f(q)$, and Schur-concave if $p \preccurlyeq q \implies f(q) \leq f(p)$.*

**Lemma 1.** *The matrix-based entropy, as given in Equation 1, is a Schur-concave function for $\alpha > 0$. This result is well-known and, for instance, was recently given by Lemma 4.1 in (Giraldo et al., 2014).*

**Theorem 4.** *Suppose we have a matrix of embeddings $Z \in \mathbb{R}^{N \times D}$ and its covariance $Z^T Z$. Then the effective rank of $Z$ is an lower bound of $\exp(S_1(Z))$, where $S_1$ denotes the matrix-based entropy of $\alpha = 1$.*

*Proof.* Denote the ordered singular values of $Z$ as $\sigma_1 \geq \cdots \geq \sigma_{\min(N,D)} \geq 0$ and the ordered eigenvalues of $Z^T Z$ as $\lambda_1 \geq \cdots \geq \lambda_{\min(N,D)} \geq 0$. Without loss of generality, assume that $\sum_{i=1}^{N} \sigma_i = \sum_{i=1}^{N} \lambda_i = 1$. If this is not the case, then set $\sigma_i := \frac{\sigma_i}{\sum_{i=1}^{N} \sigma_i}$ and $\lambda_i := \frac{\lambda_i}{\sum_{i=1}^{N} \lambda_i}$.

It is straightforward to show that $\sigma_i^2 = \lambda_i$. Because $\forall i \quad \sigma_i \leq 1$, we have that $\sigma_i \geq \lambda_i$. This implies that $\lambda \preccurlyeq \sigma$. Therefore, $S_1(\sigma) \leq S_1(\lambda) \implies$ effective rank$(Z) \leq \exp S_1(Z)$.

$\square$

**Proposition 1.** *(**Random Unit Vectors are Nearly Orthogonal**) Suppose we have $m$ unit vectors in $\mathbb{R}^D$, that are distributed according to the uniform distribution on the hypersphere. Then with probability at least $1 - m^2 \sqrt{2\pi} e^{\frac{-D\epsilon^2}{2}}$, we have that for any pair $i, j$, $i \neq j$,*

$$|\langle \mathbf{v_i}, \mathbf{v_j} \rangle| \leq \epsilon.$$

*Proof.* We can begin by defining the central $\epsilon$-band around a slice of the hypersphere $\mathbb{S}_{D-1}$ as,

$$T_\epsilon = \{z \in \mathbb{S}_{D-1} : |\langle z, e_1 \rangle| \leq \epsilon/2\},$$

where $e_1$ denotes the first basis vector. The probability of a uniformly distributed vector on the unit sphere not landing in $T_\epsilon \subset \mathbb{S}_{D-1}$ can be bounded as,

$$\mathbb{P}(T_\epsilon) \geq 1 - \sqrt{2\pi} e^{\frac{-D\epsilon^2}{2}}.$$

Now, treating $\mathbf{v_i}$ as $e_1$, the basis vector, without loss of generality, we have that, when $\mathbf{v_i}, \mathbf{v_j}$ are uniformly distributed on the hyper-sphere,

$$\mathbb{P}(|\langle \mathbf{v_i}, \mathbf{v_j} \rangle| < \epsilon) \leq \sqrt{2\pi} e^{\frac{-D\epsilon^2}{2}}$$

Now, by the union bound on each $i \neq j$, we get that,

$$\mathbb{P}(\exists i, j : \langle |\mathbf{v_i}, \mathbf{v_j} \rangle| > \epsilon) \leq \sum_{i \neq j} \mathbb{P}(|\langle \mathbf{v_i}, \mathbf{v_j} \rangle| > \epsilon)$$

$$\leq m^2 \sqrt{2\pi} e^{\frac{-D\epsilon^2}{2}}.$$

So then with probability at least $1 - m^2\sqrt{2\pi}e^{\frac{-D\epsilon^2}{2}}$, we have that, for any pair $i, j$,

$$|\langle \mathbf{v_i}, \mathbf{v_j}\rangle| \leq \epsilon.$$

$\square$

**Theorem 5.** *(**Maximum Prompt Entropy implies Large Dataset Entropy.**) Suppose we have a orthogonally equivariant representation model $Z$ such that for all sequences $Z_i = Z(X_i)$ the prompt entropy is maximal and the rows are unit. Suppose also that the data distribution* **Data** *is a isotropic unit Gaussian. Suppose we draw sequences of length $L = D$ from the data distribution. Then with probability $1 - N^2\sqrt{2\pi}e^{\frac{-D\epsilon^2}{2N^2}}$ over draw of $\{\mathbf{x_i}\}_{i=1}^N \sim$ **Data**, we have that,*

$$|e^{-S_2(QQ^\top)} - \frac{N}{L^2}| \leq \epsilon$$

*Proof.* First note that, since the prompt entropy is maximal for each sample $i$, which we denote $Z_i = Z(X_i)$, then the matrix $K_Z = Z_i Z_i^\top$ is full rank. Since by assumption each row of $Z_i$ has unit rows, then we know that $\|Z_i\|_F^2 = L = \sum_{k=1}^L \sigma_k^2$. In particular we also know that $\sigma_l = \sigma_j$ for all pairs $l, j$ by the assumption that the prompt entropy is maximized. In particular we then know that $Z_i Z_i^\top$ is a orthogonal matrix, and the rows of $Z_i$ form an orthonormal set. We can then write, for some $O_i$ a rotation matrix, that,

$$\mathbf{q_i} = \frac{1}{L}\sum_{i=1}^L \mathbf{z_i} = \frac{1}{L}O_i\mathbf{1}.$$

We will denote the average over sequences of length $L$, across all $N$ samples, by the dataset matrix $\bar{Z} = (\mathbf{q}_1, \mathbf{q}_2, \ldots \mathbf{q}_N)^\top$. Since by assumption our model $Z(\cdot)$ is orthogonally equivariant, and the **Data** distribution is radially symmetric, it follows that these $\{\mathbf{q_i}\}_{i=1}^N$ are random points on the hypersphere of radius $\frac{1}{\sqrt{L}}$. This means that the matrix $\sqrt{D}\bar{Z}$ consists of rows that are uniform points on hypersphere of radius 1. Now notice that,

$$\|\bar{Z}\bar{Z}^\top\|_F^2 = \frac{1}{L^2}\|L\bar{Z}\bar{Z}^\top\|_F^2$$
$$= \frac{1}{L^2}(\sum_{i=1}^N \|\sqrt{L}q_i\|^2 + \sum_{i\neq j}\langle\sqrt{L}q_i, \sqrt{L}q_j\rangle).$$

Since $\sqrt{L}q_i$ is a unit vector this will simplify to,

$$\|\bar{Z}\bar{Z}^\top\|_F^2 = \frac{1}{L^2}(N + \sum_{i\neq j}\langle\sqrt{L}q_i, \sqrt{L}q_j\rangle).$$

Now notice that by proposition, we have that with probability at least $1 - N^2\sqrt{2\pi}e^{\frac{-D\epsilon^2}{2N^2}}$,

$$\forall i \neq j : \langle\mathbf{v_i}, \mathbf{v_j}\rangle \leq \frac{\epsilon}{N}.$$

The union bound then tells us that,

$$\mathbb{P}(\forall i \neq j : |\langle\sqrt{D}q_i, \sqrt{D}q_j\rangle| \leq \frac{\epsilon}{N^2}) \geq 1 - N^2\sqrt{2\pi}e^{\frac{-D\epsilon^2}{2N^2}}.$$

So then with probability at least $1 - N^2\sqrt{2\pi}e^{\frac{-D\epsilon^2}{2N^2}}$ over the draw of the data points, we have that,

$$|\|\bar{Z}\bar{Z}^\top\|_F^2 - \frac{N}{L^2}| \leq \epsilon.$$

So then since,

$$S_2(\bar{Z}\bar{Z}^\top) = \log\left(\frac{1}{\|\bar{Z}\bar{Z}^\top\|_F^2}\right),$$

we have that, $e^{-S_2(\bar{Z}\bar{Z}^\top)} = \|\bar{Z}\bar{Z}^\top\|_F^2$. In particular,

$$|e^{-S_2(\bar{Z}\bar{Z}^\top)} - \frac{N}{L^2}| \leq \epsilon.$$

Which completes the proof. $\square$

**Theorem 6.** *(**Minimal Prompt Entropy Implies Small Dataset Entropy.**) Suppose we have a orthogonally equivariant representation model $Z$ such that for all sequences $Z_i = Z(X_i)$ the prompt entropy is minimal and the rows are unit. Suppose also that the data distribution* **Data** *is a isotropic unit Gaussian. Suppose we draw sequences from the data distribution. Then with probability $1 - N^2\sqrt{2\pi}e^{\frac{-D^3\epsilon^2}{2N^8}}$ over the draw of $\{\mathbf{x_i}\}_{i=1}^N \sim$ **Data**, we have that,*

$$|e^{-S_2(\bar{Z}\bar{Z}^\top)} - \frac{N^3}{L^2}| \leq \epsilon.$$

*Proof.* Since the prompt entropy is minimal for each sample, we know that each $Z(X_i)$ will be a rank one matrix, so we can write it as the outer product. In particular, we can write $Z(X_i) = \mathbf{v_i}\mathbf{u_i}^\top$. However, since the rows of $Z(X_i)$ are of unit length, we know that all the rows are identical, so we may write without loss of generality, $Z(X_i) = \mathbf{v_i}\mathbf{1}^\top$. Then, it follows that,

$$\mathbf{q_i} = \frac{1}{L}\sum_{i=j}^L \mathbf{z_j^i} = \frac{N}{L}\mathbf{v_i}.$$

We will write the dataset average matrix as before as $\bar{Z} = (\mathbf{q}_1, \mathbf{q}_2, \ldots \mathbf{q}_N)^\top$. In particular the matrix $\frac{D}{N}\bar{Z}$ has rows that are all unit vectors, and these are randomly distributed uniformly on the hyper-sphere. Now notice that,

$$\|\bar{Z}\bar{Z}^\top\|_F^2 = \sum_{i=1}^N \|\mathbf{q}_i\|^2 + \sum_{i\neq j}\langle\mathbf{q}_i, \mathbf{q}_j\rangle$$
$$= \sum_{i=1}^N \frac{N^2}{L^2}\|\mathbf{v}_i\|^2 + \sum_{i\neq j}\frac{N^2}{L^2}\langle\mathbf{v}_i, \mathbf{v}_j\rangle$$
$$= \frac{N^3}{L^2} + \sum_{i\neq j}\frac{N^2}{L^2}\langle\mathbf{v}_i, \mathbf{v}_j\rangle.$$

Now by the prior proposition, with probability at least $1 - N^2\sqrt{2\pi}e^{\frac{-D^3\epsilon^2}{2N^8}}$, we know that, for all $i \neq j$,

$$|\langle \mathbf{v_i}, \mathbf{v_j}\rangle| \leq \frac{\epsilon L^2}{N^4}.$$

So then we have that,

$$|\|\bar{Z}\bar{Z}^\top\|_F^2 - \frac{N^3}{L^2}| \leq \sum_{i \neq j} \frac{N^2}{L^2}\langle|\mathbf{v_i}, \mathbf{v_j}\rangle| \leq \frac{1}{L^2}\sum_{i \neq j}\epsilon \leq \epsilon.$$

In particular,

$$|e^{-S_2(\bar{Z}\bar{Z}^\top)} - \frac{N^3}{L^2}| \leq \epsilon.$$

$\square$

**Theorem 7.** *(Dataset Entropy Bounds InfoNCE) Let $X \sim$ **Data** be a discrete random variable distributed according to the data distribution. Let $X \to Z$ be the Markovian relation between $X$ and the representation $Z$. Then, the InfoNCE loss on $N$ samples from **Data** satisfies,*

$$\log(N) - \textit{InfoNCE} \leq I(X; Z) \leq H(Z).$$

*The entropy $H(Z)$ is analogous to the Dataset Entropy.*

*Proof.* The first inequality follows as a simple result from (Oord et al., 2018). Then, use that,

$$I(X; Z) = H(Z) - H(Z|X) \leq H(Z).$$

$\square$

# H. Additional Plots & Visualizations

Figure 8: **Relationships between representation metrics and task performance averaged across layers for Pythia 410M.** Using a variety of linear and non-linear measures—Spearman's $\rho$, Kendall's $\tau$, and distance correlation (dCor)—we see strong inversely associative relationships with the exception of InfoNCE which shows a positive, but still strong associativity. Ranges of $\rho, \tau \in [-1, 1]$ and dCor $\in [0, 1]$ with 0 indicating independence and 1 indicating strong dependency.

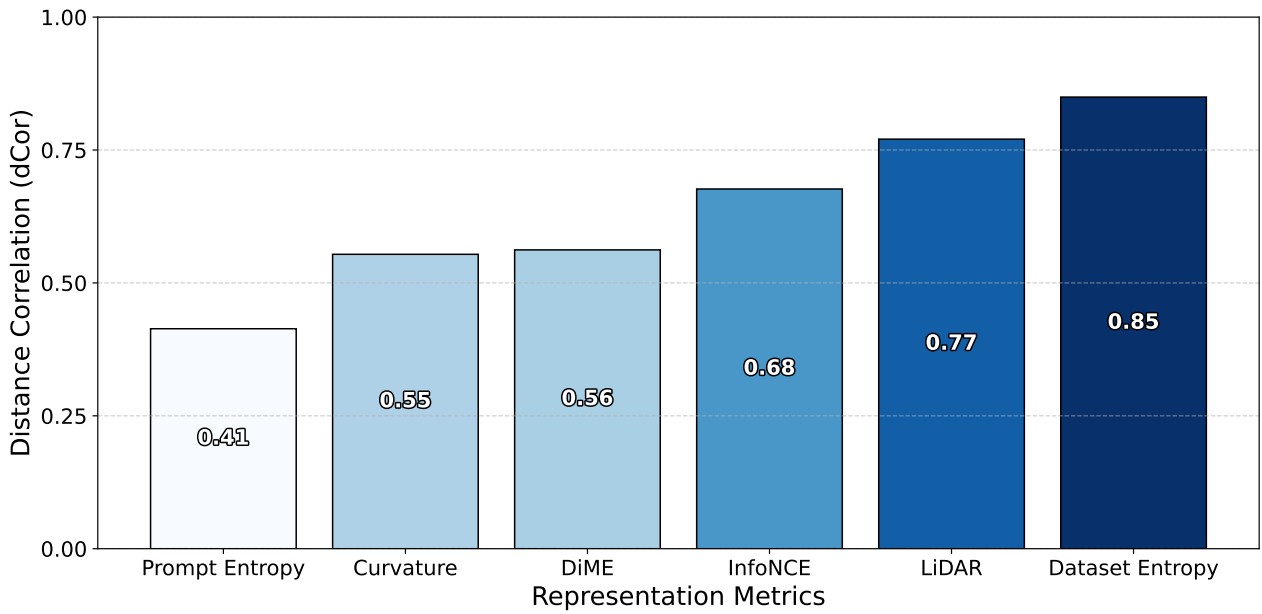

Figure 9: **Relationship between representation metrics and task performance averaged across layers for BERT.** Using distance correlation (dCor), we see strong associative relationships across the board with LiDAR and dataset entropy exhibiting the strongest relationship with downstream performance. We use dcor due to its robustness and ability to measure both linear and non-linear relationships (dCor $\in [0, 1]$ with 0 indicating statistical independence and 1 indicating strong dependency). Other correlative measures also indicate moderate to strong relationships.

| Task Domain | Tasks | # Tasks (32 Total) |
|---|---|---|
| Pair Classification | SprintDuplicateQuestions, TwitterSemEval2015, TwitterURLCorpus | 3 |
| Classification | AmazonCounterfactualClassification, AmazonReviewsClassification, Banking77Classification, EmotionClassification, MTOPDomainClassification, MTOPIntentClassification, MassiveIntentClassification, MassiveScenarioClassification, ToxicConversationsClassification, TweetSentimentExtractionClassification | 10 |
| Clustering | ArxivClusteringS2S, BiorxivClusteringS2S, MedrxivClusteringS2S, RedditClustering, StackExchangeClustering, TwentyNewsgroupsClustering | 6 |
| Reranking | AskUbuntuDupQuestions, MindSmallReranking, SciDocsRR, StackOverflowDupQuestions | 4 |
| Sentence to Sentence | BIOSSES, SICK-R, STS12, STS13, STS14, STS15, STS16, STS17, STSBenchmark | 9 |

Table 1: MTEB Tasks used in experiments covering a wide range of different use-cases and domains.

| Model | Supervised (Best) | Unsupervised | | | |
|---|---|---|---|---|---|
| | | Naive (Last) | min-DiME | min-infoNCE | min-Dataset Entropy |
| Pythia-410M | 52.0 | 45.5 | **48.5** | 46.2 | 48.1 |
| LLM2Vec-8B | 66.3 | 63.9 | 60.0 | **64.3** | 50.4 |

Table 2: Average MTEB Performance (%) across Different Layer Selection Schemes

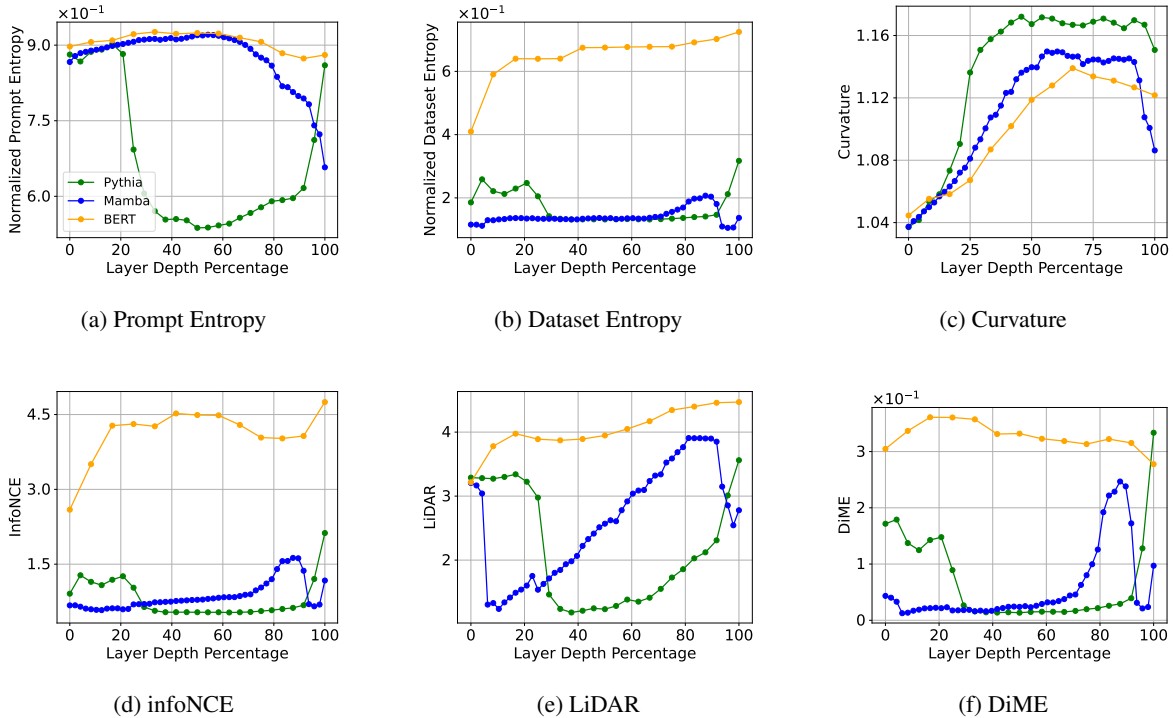

Figure 10: textbfPythia's intermediate layers show pronounced changes in representation quality metrics, while Mamba's remain more stable. Representation evaluation metrics across layers in Pythia 410M and Mamba 370M architectures. The x-axis denotes model depth as a percentage, allowing fair comparison between models with different layer counts.

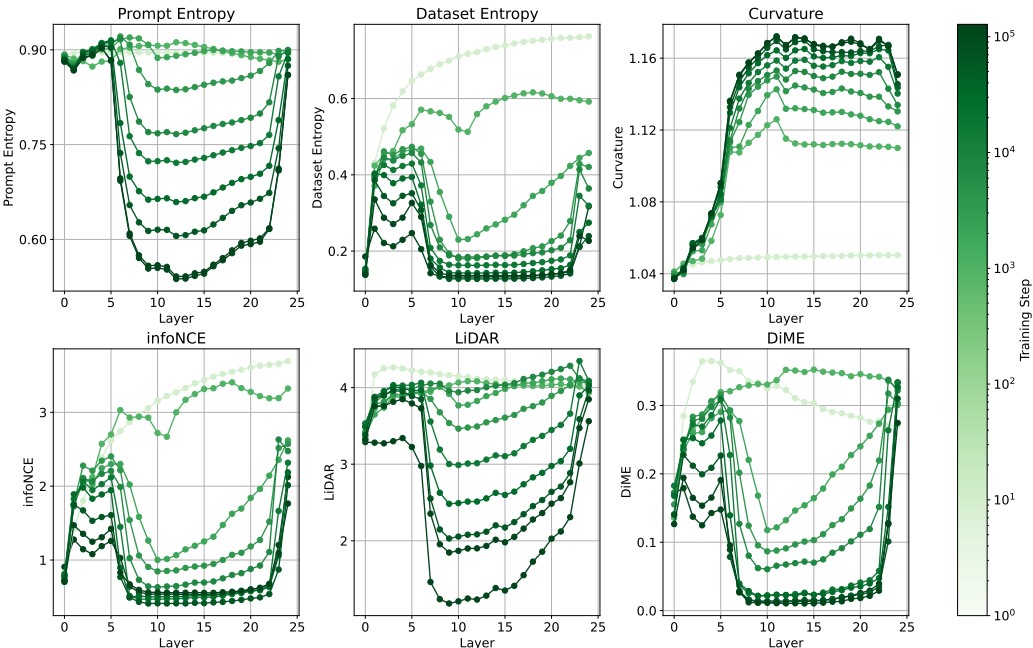

Figure 11: Representation evaluation metrics across layers at various training checkpoints, ranging from step 1 to the final step at 143k. The x-axis represents the depth percentage of the model, showing how training affects different layers, particularly in the intermediate stages.

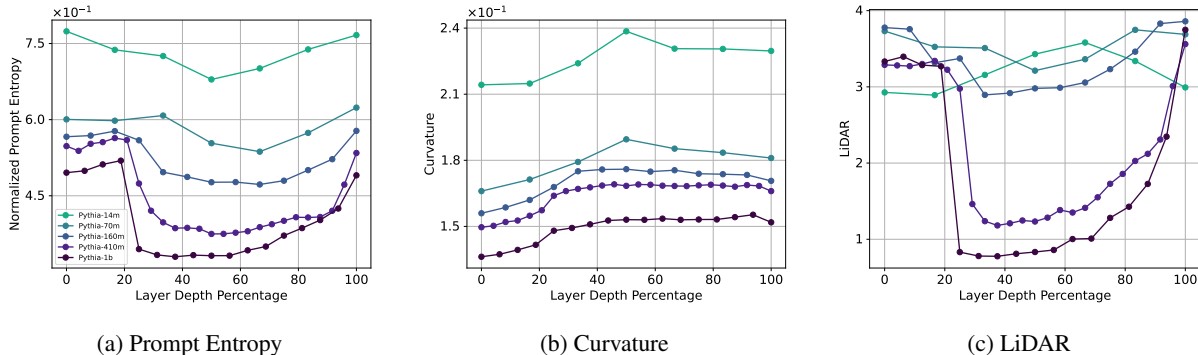

(a) Prompt Entropy        (b) Curvature        (c) LiDAR

Figure 12: **Pythia and Mamba's intermediate layers show pronounced changes in representation quality metrics, while BERT's remain more stable.** Three representation evaluation metrics calculated on the wikitext dataset for every layer in Pythia-410M, Mamba 370M, and BERT-base architectures. The x-axis denotes layer depth as a percentage, allowing fair comparison between models with different layer counts.

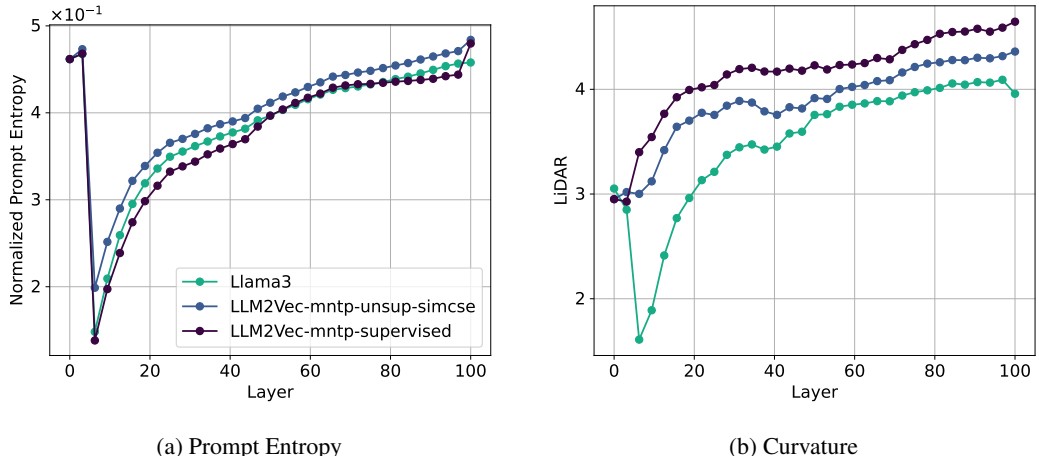

(a) Prompt Entropy                    (b) Curvature

Figure 13: **Finetuning affects the internal behavior of LLMs.** Representation evaluation metrics across layers for Llama3 and two finetuned versions of Llama3.

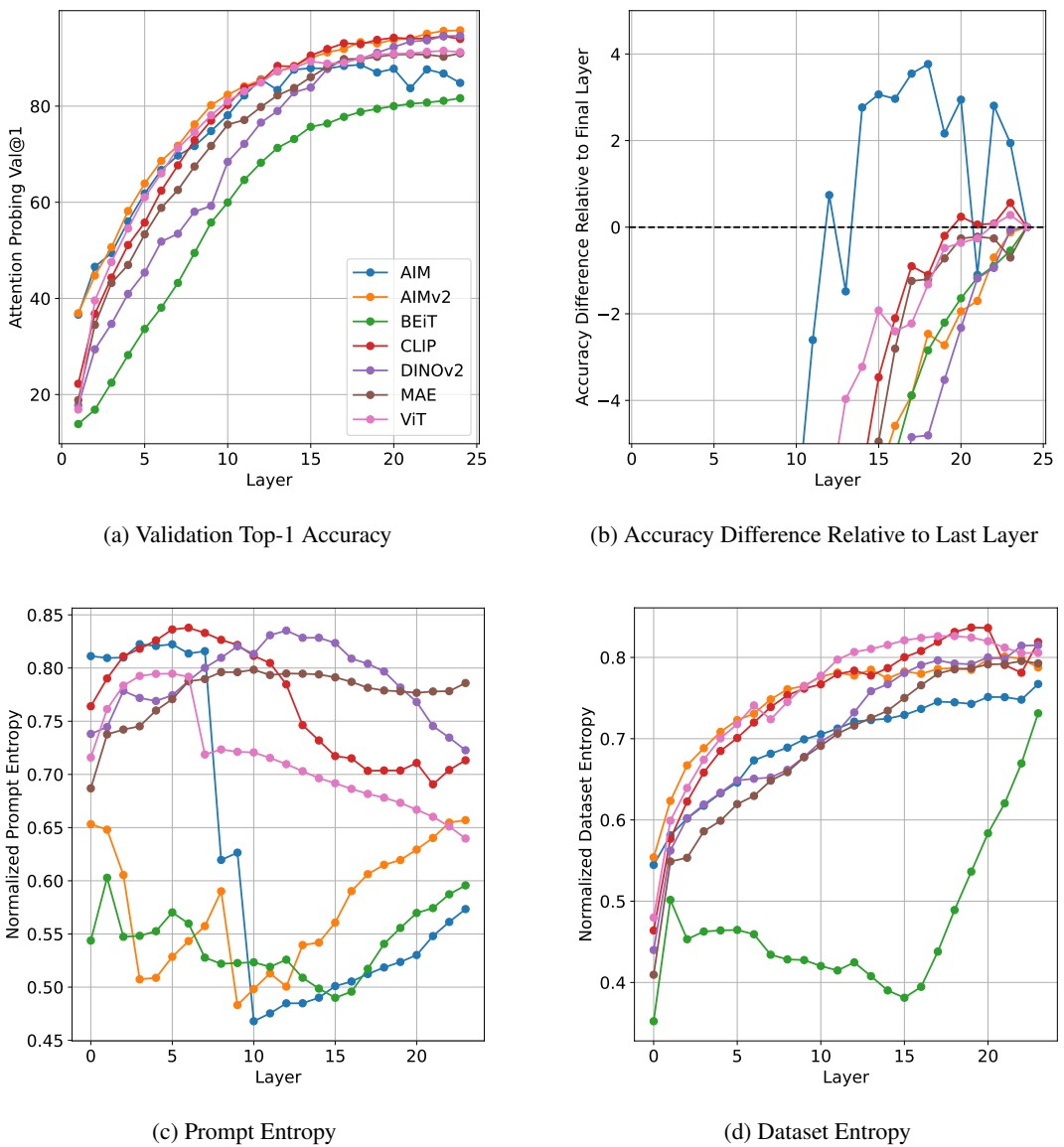

(a) Validation Top-1 Accuracy

(b) Accuracy Difference Relative to Last Layer

(c) Prompt Entropy

(d) Dataset Entropy

Figure 14: **Comparison of vision models trained on different pretext tasks**. The dataset is ImageNet-100 (Tian et al., 2020) and all models use the same 24-layer ViT-L architecture. The validation accuracy is calculated using attention probing on tokens from a frozen backbone layer, following the work of El-Nouby et al. (2024)

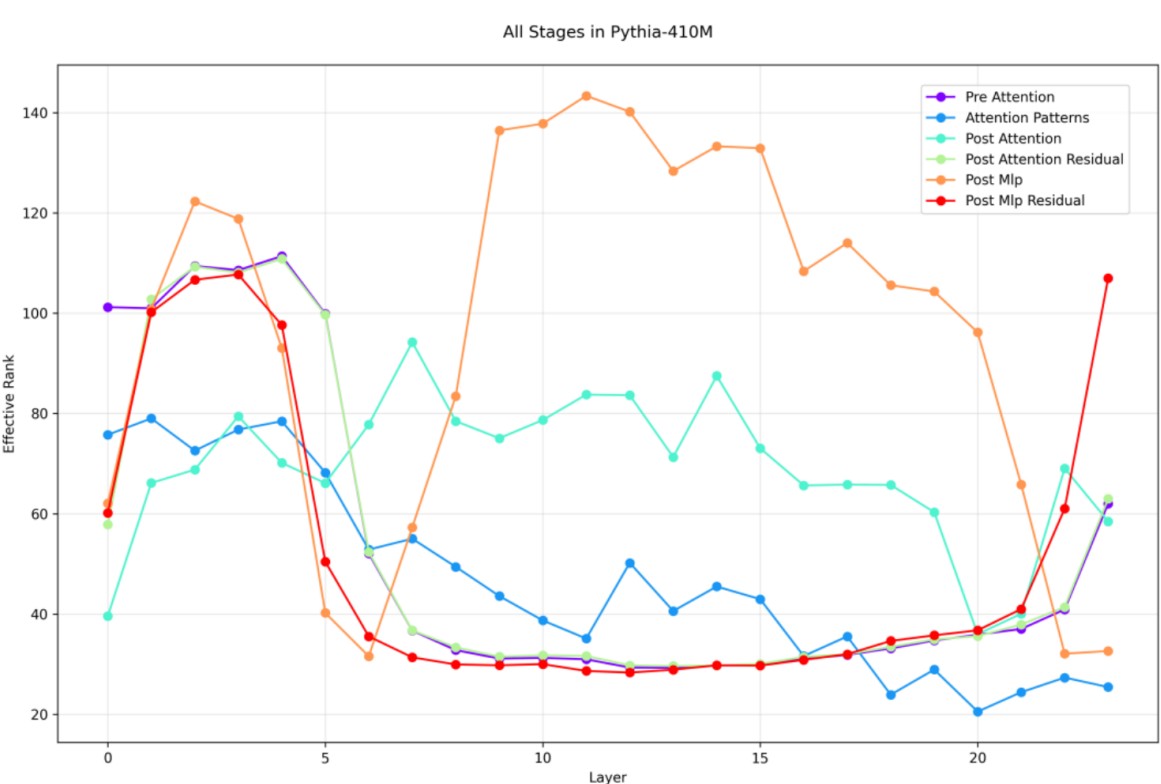

Figure 15: Behavior of effective rank at different stages within a transformer block.

