# OpenReview forum: "Layer by Layer: Uncovering Hidden Representations in Language Models"
_ICML.cc/2025/Conference — ICML 2025 oral_

### Official Review · Reviewer_U8pM · 2025-02-18

**Overall Recommendation:** 3

**Summary:**

The paper proposes a framework for analyzing representations throughout model layers. From the perspective of matrix-based entropy, they primarily measure properties like compression (e.g., prompt entropy), geometric smoothness (e.g., curvature), and augmentation invariance (e.g., LiDAR). They argue that for autoregressive models, intermediate layers outperform late ones, unlike bidirectional models. They demonstrate a similar trend for vision models. They also investigate trends related to model size, training progression, and chain-of-thought models.

**Claims And Evidence:**

- Claim 1. There is a theoretical perspective using matrix-based entropy that unifies many metrics.
    - While Sec. 3.4 presents the theoretical connections between matrix-based entropy and prompt and dataset entropy and InfoNCE, these results feel disconnected from the rest of the paper. Namely, dataset entropy and InfoNCE are not tested or discussed in the empirical experiments of the main text.
    - Only a subset of metrics discussed in Sec. 3.3 appear in the main paper, while the full set appears in Figure 8 of the Supplementary. It would be nice to include a summary of these results in the main paper.
- Claim 2. The proposed framework reveals why intermediate layers outperform late ones.
    - The paper does a reasonable job of validating this claim in Sec. 4.2 and Sec. 5, although the presentation could be improved.
    - The discussion in L302 is too brief. Figures 1 and 2 indeed shows a relationship between trends in performance and prompt entropy, which can be used to contrast autoregressive (Pythia) and bidirectional models (BERT). However, the curvature and LiDAR plots are not discussed. The interpretation in L317 is incomplete — Mamba exhibits conflicting trends across metrics (in prompt entropy at 80% depth the layers are some of the most compressed, in LiDAR they are the least compressed). The result in Figure 1 where BERT’s best representation is at 10% depth should be discussed; the “conventional wisdom” would suggest that the last representation would perform the best, but that is not the case there.
    - L408 / Figure 13 is missing a key performance result. In Figure 13, it is unclear why AIM is not plotted. To complete the argument, the paper should validate for autoregressive vision models, whether an intermediate layer performs the best as well.
- Other Claims
    - Compression increases as models scale. This is shown in Figure 10; I liked this result and it was clearly conveyed.
    - Compression increases with more training steps. This is shown in Figure 4; I again thought this was a nice result that was clearly conveyed.
    - Residual connections drive compression. I am not sure I am fully convinced; L346 could be more thoroughly and clearly explained. I would have liked a more detailed explanation on the setup and annotations for the Figure 15 legend (e.g., explaining “pre attention,” “attention patterns,” etc.), and a longer walkthrough on pairings in the figure (e.g., compare the “Post Mlp” and “Post Mlp Residual”) to better understand this analysis.

**Essential References Not Discussed:**

N/A

**Experimental Designs Or Analyses:**

See “Claims and Evidence” above.

**Methods And Evaluation Criteria:**

See “Claims and Evidence” above.

**Other Comments Or Suggestions:**

In Figure 2 and other similar figures, it would be helpful to include some more intuitive semantic label for each metric. I have to spend some effort recalling that lower entropy means more compression, higher curvature means more abrupt changes, etc.

**Other Strengths And Weaknesses:**

Strengths

- The presentation of matrix-based entropy in Sec. 3.2 is well-motivated and conveyed in an intuitive fashion.
- The paper evaluates not only on language models but also vision models, demonstrating the generality of its findings.
- The paper reveals compelling findings like larger models and more training steps lead to more representational compression.

Weaknesses

- The paper is missing some important discussion of state space models and autoregressive vision models (see “Claims And Evidence” above).
- The paper mentions a number of metrics in its unified framework and core theoretical results in Sec. 3, yet only a subset is discussed in the main text. The presentation could be improved with a more clear connection between the theoretical and empirical results (see “Claims And Evidence” above).

I am open to updating my score if the authors are able to address the listed weaknesses.

**Questions For Authors:**

N/A

**Relation To Broader Scientific Literature:**

The framework of analysis is the main novel contribution of this paper. The paper correlates performance and various pre-existing compression metrics, for autoregressive vs. bidirectional models, and other settings.

**Theoretical Claims:**

N/A

---

> ### Author Rebuttal · Authors · 2025-04-01
>
> We sincerely thank Reviewer U8pM for the detailed feedback and for indicating openness to reconsidering the evaluation. We appreciate the positive comments on our work. We address your specific points below. Note, that You may also be interested in the new experimental results provided for Reviewers 9n2s (generative tasks) and ctdG (unsupervised mechanism for selecting good intermediate layers).
>
> ## Addressing Framework Metrics and Theory-Empirical Connection (Claim 1 / Weakness 2)
>
> You correctly noted that while Sec 3 outlined a framework connecting several metrics (including Dataset Entropy, InfoNCE) via theory, the main empirical discussion focused on a subset. This was primarily due to space constraints. However, we intend to take advantage of the extra page in the final submission to do exactly as you suggest and make a longer discussion. We will utilize the additional page in the camera-ready version to:
>
>  - Explicitly discuss the empirical results for **all metrics shown in Appendix Figure 8** in the main text.
> -  Expand the discussion in Sec 4.2 to fully incorporate Curvature and LiDAR trends alongside Prompt Entropy when comparing model architectures (Pythia, BERT, Mamba), addressing the noted brevity in L302.
>
>
> ## Addressing Empirical Validation and Presentation (Claim 2 / Weaknesses 1 & 2)
>
> ***Missing AIM Performance / AR Vision Validation:*** Thank you for pointing out the missing AIM results in Figure 13. This validation is indeed key. We have now performed this analysis and present the results for AIM (and BEiT) linear probe accuracy layer-wise in Table 3 below.
>
> As shown, the autoregressive vision model AIM exhibits a modest performance gain (+1.9\%) at an intermediate layer (75\% depth) compared to the final layer. This aligns with our findings in language models and provides the requested validation. BEiT, consistent with other non-AR models, peaks at the final layer. We will integrate these results into Figure 13 and the main text discussion.
>
> ### Table 3: Val@5 Linear Probe Accuracy on ImageNet-1k at Different Layer Depths
>
> | Model | 0%   | 25%   | 50%   | 75%                | 100% (Final) |
> |--------|:-----|:------|:------|:-------------------|:-------------|
> | AIM    | 3.8% | 13.7% | 28.5% | 82.0% (L18)       | 80.1%        |
> | BEiT   | 2.9% | 7.1%  | 14.6% | 46.8%             | 62.5%        |
>
>
> **Mamba Discussion (Conflicting Metrics)** Regarding the observation at 80\% depth: Prompt Entropy indicates high compression, while LiDAR indicates high augmentation invariance. These are not conflicting, but rather co-occurring, properties. Thus, the interpretation is: "Mamba's most compressed layers (via entropy) are also its most augmentation invariant (via LiDAR)". We will clarify this distinction and provide a more detailed discussion of Mamba's unique trends in the expanded Sec 4.2. We will also include a discussion of BERT, which we omit here for character limits.
>
> ## Addressing Residual Connection Claim (Claim 2)
>
> We appreciate the request for clarification on the "Residual connections drive compression" claim.
>
>
> **Figure 15 Legend Details:** To clarify the setup, we used hooks to capture activations at key stages within each transformer block:
>
> - **Initial Representations** Input to the block's residual stream.
> - **Attention Patterns** Raw attention weights (query-key interactions).
> - **Attention Outputs** Output of the attention mechanism value aggregation, projected back.
> - **Attention Residuals** Result after adding attention output to the initial residual stream and applying LayerNorm.
> - **MLP Output** Output of the MLP layers, projected back.
> - **MLP Output + Residuals** Final block output (input to next layer), after adding MLP output to the **Attention Residuals** stream. This is the typical "layer output" we measure.
>
>
> **Explanation**
> Comparing MLP Output (step 5) with MLP Output + Residuals (step 6) in Figure 15 reveals the latter has a significantly lower effective rank (is more compressed). This occurs because the norm of the residual stream (Attention Residuals, step 4) is often much larger than the norm of the MLP Output (step 5). When added together, the high-norm residual component dominates the sum, effectively reducing the combined representation's rank/increasing compression compared to the MLP output alone. We will incorporate this detailed walkthrough into Sec 4 and add plots **showing the norms** of these components alongside Figure 15 to make this argument clearer and more convincing.
>
> ## Conclusion
> We hope these responses, clarifications, and new results address the reviewer's points. We will incorporate these changes into the camera-ready version. Thank you again for the feedback. We hope these comprehensive responses and improvements demonstrate the soundness and significance of our work. We would be grateful if you would consider these points in reassessing our submission, and we welcome any further questions.

---

> > ### Comment · Reviewer_U8pM · 2025-04-01
> >
> > - I reviewed the missing experiment on AIM; the provided table indeed shows a peak in accuracy at an intermediate layer, in contrast with the non-autoregressive models.
> > - Thank you for the clarified discussion of the Mamba results. I would also encourage the authors to address this observation about BERT: *in Figure 1 [...] BERT’s best representation is at 10% depth.* However, this point does not affect my score.
> > - Thank you for the detailed walkthrough of the results in Figure 15. It is an intriguing explanation that the residual stream is behaving as a noise filter, where it dominates the representation with a higher norm.
> >
> > Given the clarifications, I have increased my score.

---

> > > ### Author Response · Authors · 2025-04-04
> > >
> > > Thank you again for your review, and we are glad that our clarifications addressed your concerns. We are grateful that you have raised your score of the paper.

---

### Official Review · Reviewer_aJrj · 2025-03-13

**Overall Recommendation:** 4

**Summary:**

This paper introduces a unified framework for evaluating representation quality in language models. The framework is based on information theory, geometry, and invariance to input perturbations. The authors analyze how each layer balances information compression and signal preservation, challenging conventional wisdom by showing that intermediate layers outperform the final layers in certain tasks.

The proposed method is tested across a diverse set of 32 text-embedding tasks, covering various model architectures and domains. The authors also explore when and why these findings hold by linking them to fundamental properties such as entropy, invariance, and geometry via matrix-based entropy.

**Claims And Evidence:**

The main aim of this work is to propose a unified framework for evaluating the representations of layers in LLMs. One of the initial claims is that intermediate layers can surpass final layers in performance due to their superior balance of compression and signal retention.

To this end, this study provided both a theoretical analysis of the matrix based entropy and an empirical validation using a broad set of tasks and models that seem to confirm the claims of the authors.

**Essential References Not Discussed:**

In my view, the most essential works are appropriately discussed.

**Experimental Designs Or Analyses:**

The experiments cover an extensive range of 32 text-embedding tasks, but additional details on how the prompts were constructed and how many were used per experiment,  would help clarify the settings and improve reproducibility. Are these part of the considered datasets?

**Methods And Evaluation Criteria:**

The study evaluates representation quality using metrics derived from information theory, geometry, and robustness to perturbations. Various  measures are used in this context based on nominal works such as the InfoNCE, while considering not only different tasks but also architectural families of models.

To the best of my knowledge, even though the use of matrix-based entropy is not novel in the community, the interpretation of the arising properties combining the three different information theoretic, geometric and invariance properties, along with an empirical extensive evaluation, is.

**Other Comments Or Suggestions:**

No further comments.

**Other Strengths And Weaknesses:**

No further comments.

**Questions For Authors:**

Most of my concerns/questions have been expressed in the previous sections.

**Relation To Broader Scientific Literature:**

I find that the proposed work is largely based on the previous works of (Giraldo et al., 2014; Skean et al., 2023) that are properly cited.
Even though this study does not constitute a groundbreaking contribution per se, it does provide a unified framework, concentrating and evaluating the results of previous works.

**Theoretical Claims:**

I check the theoretical claims of the paper. I would like some clarifications about the appropriate conditions on the data distribution and model that is required for Theorem 2, and if these hold in practice.

I would  also like some clarifications on the prompt entropy and how we can construct prompts that are appropriate in the context of the provided analysis.

---

> ### Author Rebuttal · Authors · 2025-04-01
>
> Thank you for your thoughtful comments and feedback. We appreciate you recognizing the novelty in our unified interpretation and extensive evaluation. You may also find our new results relevant (details in responses to Reviewers 9n2s regarding generative tasks and ctdG regarding unsupervised layer selection).We address your specific points below:
>
> ### Clarification on Theorem 2 Conditions
>
> You asked about the conditions required for Theorem 2 and their applicability in practical scenarios. Our theoretical framework relies on three main assumptions:
>
> -  **Orthogonal Equivariance:** We assume an orthogonally equivariant model. While transformers are generally permutation equivariant, this stronger condition enables theoretical tractability, though we acknowledge it's a simplification for standard models.
>
> - **Gaussian Input Data:** We assume input data follows a Gaussian distribution, which simplifies analysis due to its tractable mathematical properties.
>
> - **Representations on Hypersphere:** We assume token representations lie on a hypersphere. This is often approximated in practice due to Layer Normalization in high dimensions, which yields representations with nearly constant norms.
>
>
> We believe these assumptions collectively provide a reasonable, albeit simplified, foundation for analyzing modern transformers. We will **explicitly state these assumptions** in the revised theorem statements and add a **discussion in the main text** outlining these conditions and the practical scenarios where deviations might occur.
>
> ### Clarification on Prompt Construction
>
> Regarding your question on prompt construction, we give the following explanation and example. The Massive Text Embedding Benchmark (MTEB) framework we use provides standardized code to automatically generate prompts for each task. We used these default MTEB prompts without modification to ensure consistency and reproducibility. We recognize that examples were missing in our submission. We will add a discussion of the MTEB prompting strategy and include concrete examples (like the one below for EmotionClassification) in Appendix D.2.
>
> - **Example Prompt Format (EmotionClassification)** "Classify the emotion expressed in the given Twitter message into one of the six emotions: anger, fear, joy, love, sadness, and surprise: {{SAMPLE GOES HERE}}"-
> - **Example Full Prompt:** "Classify the emotion expressed in the given Twitter message into one of the six emotions: anger, fear, joy, love, sadness, and surprise: Thank you, reviewer!"
>
>
> ### Prompt Entropy
> You also asked for clarification on prompt entropy. We hope the explanation above clarifies the prompt construction aspect. Regarding prompt entropy itself (our metric measuring token-level information uniformity), we are happy to elaborate further on its calculation or interpretation if specific aspects remain unclear.
>
>
> ### Conclusion
> We have worked to address your points, particularly regarding the assumptions underlying our theory and the details of prompt construction. We hope these clarifications strengthen the paper in your view. Given these explanations and our commitment to updating the manuscript accordingly, we would be grateful if you might reconsider your assessment. Please let us know if any further questions or points require clarification.

---

### Official Review · Reviewer_9n2S · 2025-03-14

**Overall Recommendation:** 3

**Summary:**

This study investigates whether intermediate layers of language models offer more informative representations compared to the final layers. The authors propose a unified framework for assessing representation quality, employing seven evaluation metrics categorized into three groups:  information-theoretic, geometric, and augmentation-invariance. The experimental design encompasses three model architectures: decoder-only transformers, state space models, and encoder-only transformers. The study evaluates each layer's embeddings across 32 tasks from the MTEB benchmark, covering classification, clustering, and reranking. Through comprehensive analysis, the research examines the performance of intermediate layers across different architectures, explores their variations under diverse training paradigms, and investigates the consistency of findings between textual and visual models.

**Claims And Evidence:**

The claims made in this submission are supported by clear and convincing evidence. The authors conduct a comprehensive empirical investigation, encompassing diverse scenarios such as downstream task performance and parameter variations during training. Through systematic examination of layer-wise representations across various model architectures, the study shows that intermediate layers of language models can encode richer representations compared to their final layers.

**Essential References Not Discussed:**

The paper explores the performance of intermediate layers in language models, which is a topic of growing interest in the field. However, there are relevant studies addressing similar themes that have not been cited:

[1] Is Bigger and Deeper Always Better? Probing LLaMA Across Scales and Layers;

[2] Transformer Layers as Painters.

**Experimental Designs Or Analyses:**

Most of the experimental design is well-aligned with the study's objectives, establishing a robust connection between the evaluation framework and the empirical findings.

However, the experiments "Finetuning Effects" and "Impact of Chain-of-Thought Finetuning" do not appear directly relevant to the main task. The reviewer is unclear about the rationale for including these experiments.

**Methods And Evaluation Criteria:**

The proposed methods and evaluation criteria are both appropriate and well-justified. The authors conduct a comprehensive evaluation across 32 MTEB tasks, systematically assessing representative models of various architectures through three distinct perspectives: information-theoretic, geometric, and augmentation-invariance. In addition, the authors provide a thorough explanation for their selection of specific tasks and evaluation metrics.

However, while the paper conducts experiments across 32 text embedding tasks, these tasks are predominantly focused on classification, clustering, and reranking. The study does not encompass other critical NLP tasks, such as machine translation, question answering, or dialogue generation, which are essential for a more comprehensive evaluation.

**Other Comments Or Suggestions:**

(1) Present the experimental details of AIM, such as its scale, along with the corresponding results.

(2) Expand the evaluation to include tasks like machine translation, question answering, or dialogue generation.

(3) Consider extending the dataset domains to areas such as medicine and law.

**Other Strengths And Weaknesses:**

Other Strengths:
(1) The paper is well-written and well-organized, ensuring clarity for readers to grasp the core arguments and findings.

(2) The research topic is engaging and scientifically significant, with experimental results that offer valuable insights and implications for the broader research community.

(3) The experiments are comprehensive, offering substantial evidence to support the authors' claim that intermediate layers can, in certain cases, outperform the final layers.

(4) The authors have provided their codes for reproducibility.



Other Weaknesses:

(1) The experimental details and results for the AIM are not presented in the paper.

(2) While the paper employs MTEB as its primary evaluation benchmark, it is important to note that MTEB may not encompass the full spectrum of text data types or tasks. A notable limitation is the absence of low-resource languages or specialized domains such as medicine, law, and others.

(3) It will be more rigorous to discuss the limitations of the paper.

**Questions For Authors:**

(1) What is the rationale for including the "Finetuning Effects" and "Impact of Chain-of-Thought Finetuning" in the main empirical results?

(2) Are there any plans to conduct experiments on more advanced LLMs or MLLMs, such as Qwen2.5 and Janus?

**Relation To Broader Scientific Literature:**

In the field of knowledge distillation, several studies explore the knowledge transfer based on the intermediate layers, which aligns closely with the core contributions of this paper.

Relevant works in this area include:

[1] Knowledge Flow: Improve Upon Your Teachers;

[2] MergeNet: Knowledge Migration across Heterogeneous Models, Tasks, and Modalities.

**Theoretical Claims:**

I have reviewed the theorems in the Core Theoretical Results section and it seems right to me.

---

> ### Author Rebuttal · Authors · 2025-04-01
>
> We sincerely thank Reviewer 9n2S for the detailed review, positive feedback on our claims, methods, and writing, and constructive suggestions. We are encouraged by the reviewer's assessment of our work as engaging and scientifically significant.
> We address the reviewer's comments and questions below, including new experimental results motivated by the feedback. Note that they may also be interested in the new experimental results provided for Reviewers ctdG (unsupervised mechanism for selecting good intermediate layers).
>
> ## Addressing Evaluation Scope (Non-Embedding Tasks):
>
> We appreciate the reviewer highlighting the importance of evaluating beyond embedding-centric tasks. To address this, **we conducted new experiments on generative and classification tasks.** Evaluating intermediate layers for generation requires obtaining logits, typically via an "unembedding" layer. We employed **the TunedLens technique** [1], which enables layer-wise analysis of generative capabilities. These results complement our MTEB findings, showing that (a) intermediate layers can outperform final layers on diverse tasks like QA, and (b) the optimal layer depth is task-dependent.
>
>
> **New Results (QA & Classification)**
> * On MMLU (Question Answering), using Llama3-8B with TunedLens, we found **Layer 28 achieves an average accuracy of 62.1\%, outperforming the final layer's 61.2\% (+0.9\%)**. This demonstrates that intermediate layers can offer benefits beyond MTEB tasks. (See Table 2 for aggregated results).
> * We also performed layer-wise analysis on **ToxiGen** (toxicity detection) and **BLIMP** (grammaticality judgments) using LM-evaluation-harness. Interestingly, optimal performance varied: early layers excelled on ToxiGen, while late layers were best for BLIMP.
>
>
> ### Table 2: Average MMLU Accuracy with TunedLens at Different Llama3-8B Layers
>
> | Layer Depth   | 0%    | 25%   | 50%   | 75%                        | 100% (Final) |
> |---------------|:------|:------|:------|:---------------------------|:-------------|
> | MMLU Score    | 22.9% | 22.9% | 25.7% | 60.7% (L28 peak: 62.1%)   | 61.2%        |
>
> ## Responses to Questions & Other Points
>
> **1: Rationale for Finetuning/CoT Experiments:** The primary goal of these experiments was not just performance evaluation, but to demonstrate the **utility and sensitivity of our proposed Matrix Entropy framework** for probing LLM behavior under different conditions (finetuning, CoT prompting), aligning with recent work like Seq-VCR [2]. We will reorganize the paper and incorporate relevant figures (currently in the appendix) into the main text.
>
> **2: Experiments on Larger/Advanced Models:** We plan to experiment with the multimodal Llama3.2-11B-Vision model, which provides a valuable bridge between our language (Sec 4) and vision (Sec 5) experiments. Models like Qwen2.5 and Janus are excellent candidates for future work to further scale our findings.
>
> **3. AIM Plots** We apologize for omitting the AIM details and results. These are included in our response to Reviewer U8pM and will be added to the main paper in the camera-ready version.
>
> **4. Additional References** Thank you for suggesting these relevant references on knowledge distillation and layer analysis. They offer valuable perspectives on knowledge transfer and feature encoding in intermediate layers. We will incorporate and discuss these works in the related work section, strengthening our connection to the literature and helping address questions about intermediate features (raised also by Reviewer ctdG).
>
> **5. Discussion of Limitations** We acknowledge the points raised regarding limitations and scope. As discussed above, our new results broaden the task variety beyond MTEB. Regarding limitations involving theoretical assumptions, as noted by Reviewer aJrj, our framework makes certain assumptions (detailed in our response to aJrj). We will clearly outline these in the revised manuscript's discussion section.
>
>
> ## Conclusion
> We have put significant effort into addressing the points raised in your review, including conducting new experiments on non-embedding tasks (MMLU, ToxiGen, BLIMP) using the TunedLens approach, which directly addresses one of your main concerns and further demonstrates the generality of our findings. We hope these additions, along with our clarifications and planned revisions, strengthen the paper and demonstrate the value of our contributions. In light of these new results and our detailed responses, we would be grateful if you would consider raising your assessment of our work. Please do not hesitate to let us know if any further questions or concerns arise; we welcome the opportunity for continued discussion. Thank you again for the thoughtful feedback.
>
>
> ## References
> [1] Belrose et. al., "Eliciting Latent Predictions from Transformers with the Tuned Lens",  2023.
>
> [2] Arefin, et al, "Seq-VCR: Preventing Collapse in Intermediate Transformer Representations for Enhanced Reasoning", 2024

---

### Official Review · Reviewer_ctdG · 2025-03-14

**Overall Recommendation:** 4

**Summary:**

This paper investigates representation quality across different layers of large language models (LLMs), challenging the conventional wisdom that final-layer embeddings are optimal for downstream tasks. Through systematic evaluation on 32 tasks from the Massive Text Embedding Benchmark, the authors demonstrate that intermediate layers often outperform final layers by up to 16% in accuracy. They propose a unified theoretical framework integrating information-theoretic, geometric, and invariance perspectives to explain this phenomenon, showing how matrix-based entropy can measure how effectively each layer balances information compression and signal preservation. The authors compare multiple architectures (transformers, state-space models) and domains (language, vision), finding that autoregressive models exhibit a pronounced mid-layer "compression valley" while bidirectional models show more uniform patterns. Additional analyses examine how representation quality evolves during training, how chain-of-thought finetuning affects mid-layer entropy, and how similar patterns emerge in vision models with analogous training objectives.

**Claims And Evidence:**

The paper's central claims are well-supported by substantial empirical evidence. The assertion that intermediate layers outperform final layers is convincingly demonstrated through comprehensive benchmarking across diverse tasks and model architectures. The authors provide both quantitative metrics and visualizations showing how representation quality varies with depth.

The theoretical framework connecting entropy, geometry, and invariance is supported by mathematical formulations and empirical correlations between these metrics and downstream performance. The correlation analysis (Figures 3, 6, 7) demonstrating strong relationships between their proposed metrics and task performance provides compelling evidence for the validity of their framework.

The architecture-specific claims about compression patterns are well-supported by consistent findings across multiple model scales and types. The extension to vision models provides additional credibility to their argument that the observed patterns are driven by training objectives rather than data modality.

**Essential References Not Discussed:**

The authors have discussed carefully in their paper.

**Experimental Designs Or Analyses:**

The experimental designs are sound and well-executed. The layer-wise evaluation approach provides a comprehensive view of how representation quality evolves across model depth. The analysis of model architectures is systematic and controls for relevant variables by normalizing layer depths as percentages to allow fair comparison across different model sizes.

The training progression analysis (Figure 4) effectively captures how representations evolve during training, providing valuable insights into the dynamics of representation learning. The sub-component analysis of transformer blocks (Figure 15) is particularly illuminating, isolating the effects of different components (attention, MLP, residuals) on representation quality.

One limitation is that while the authors demonstrate superior performance of intermediate layers, they don't provide a systematic way to identify which specific intermediate layer is optimal for a given model or task without empirical testing.

**Methods And Evaluation Criteria:**

The evaluation methods are appropriate and comprehensive. Using the MTEB benchmark with 32 diverse tasks provides a robust test of representation quality across different use cases. The authors' approach of testing every layer systematically on each task allows for direct, fair comparisons.

The set of metrics developed to assess representation quality (prompt entropy, dataset entropy, curvature, LiDAR, DiME, InfoNCE) collectively capture different facets of what makes representations effective. The combination of these metrics with downstream performance establishes a clear connection between theoretical properties and practical utility.

The inclusion of multiple architectures (Pythia, Mamba, BERT) at various scales strengthens the generalizability of the findings. The controlled experiments with extreme input conditions (repetition, randomness, varying length) provide additional insights into how different layers process information.

**Other Comments Or Suggestions:**

- The presentation of results could be enhanced by providing concrete examples of what types of features or patterns are being captured by intermediate vs. final layers.
- It would be valuable to include a discussion of the computational implications of using intermediate layer representations rather than final layers.
- A discussion of how these findings might inform architectural design choices for future LLMs would strengthen the impact.

**Other Strengths And Weaknesses:**

**Strengths:**
- The integration of multiple perspectives (information theory, geometry, invariance) into a unified framework is elegant and insightful.
- The cross-domain validation with vision models significantly strengthens the generalizability of the findings.
- The analysis of sub-components within transformer blocks provides valuable insights into the mechanisms driving the observed patterns.
- The practical implications for representation extraction are significant and could influence how embeddings are utilized in downstream applications.

**Weaknesses:**
- The paper doesn't establish clear guidelines for selecting the optimal layer for a specific task or model without empirical testing.
- The connection between the observed patterns and the actual content being represented at different layers remains somewhat abstract.
- The paper doesn't discuss potential connections to other observed phenomena in transformers, such as the attention sink effect, which might provide additional explanatory power.
- The theoretical justification for why autoregressive models develop stronger mid-layer compression than bidirectional models could be more thoroughly developed.

**Questions For Authors:**

1. Have you investigated whether there are consistent patterns in which specific tasks benefit most from intermediate versus final layer representations? This could help develop guidelines for which layer to use for different application types.

2. Did you observe any relationship between the optimal layer depth and model scale? While you show that compression patterns become more pronounced in larger models, does this shift where the optimal layer for downstream tasks is located?

3. Your framework suggests that intermediate layers achieve an optimal balance between compression and signal preservation. Have you explored whether this balance can be explicitly optimized during training to enhance representation quality further?

**Relation To Broader Scientific Literature:**

The authors have discussed carefully in their paper.

**Theoretical Claims:**

The paper makes several theoretical claims that appear sound. I verified the connections between matrix-based entropy and effective rank (Theorem 4), which is correctly established. The relationship between InfoNCE and entropy (Theorem 7) is well-grounded in information theory.

The theorems connecting prompt entropy to dataset entropy (Theorems 5 and 6) provide useful insights into how local (token-level) and global (dataset-level) properties interact. These theoretical results help explain why certain compression patterns lead to more effective representations.

The unification of seemingly disparate metrics under a common framework of matrix-based entropy is a significant theoretical contribution that appears technically sound, though some of the proofs would benefit from more detailed derivations.

---

> ### Author Rebuttal · Authors · 2025-04-01
>
> We sincerely thank Reviewer ctdG for the detailed, insightful review and constructive feedback. We appreciate the positive assessment of our claims, methods, theory, and experiments. Below, we address the reviewer's comments and questions, incorporating clarifications and new results. You may also be interested in the new results provided for Reviewers 9n2s (generative tasks). Due to character limits, we could not fully address your suggestions about architectural and computational implications, though we will incorporate these suggestions into the manuscript.
>
> ## New Results: Systematic Way to Select Optimal Layer
> We agree guidance on unsupervised layer selection is valuable. Our correlation results showed unsupervised metrics (entropy, DiME, etc.) can act as proxies for downstream performance. To demonstrate this, **we present new results** (Table 1) evaluating unsupervised layer selection. We compare the naive last layer, the supervised best layer, and layers chosen by selecting those with minimum DiME, Dataset Entropy, or InfoNCE per task. As shown, for both Pythia-410M and LLM2Vec-8B, **specific unsupervised metrics yield better average MTEB performance than the last layer** confirming their utility. We will clarify this procedure and its benefits in the revised manuscript.
>
> ### Table 1: Average MTEB Performance with Different Layer Selection Schemes
>
> | Model        | Naive (Last) | Supervised (Best) | Unsupervised (min-DiME) | Unsupervised (min-Dataset Entropy) | Unsupervised (min-infoNCE) |
> |--------------|:------------:|:-----------------:|:------------------------:|:---------------------------------:|:---------------------------:|
> | Pythia-410M  | 45.5%        | 52.0%             | 48.5%                    | 48.1%                             | 46.2%                       |
> | LLM2Vec-8B   | 63.9%        | 66.3%             | 60.0%                    | 50.4%                             | 64.3%                       |
>
> ## Responses to Points
> **Detailed Proofs** We agree that additional detail in the derivations for Theorems 5 and 6, connecting local and global token-level properties, would be beneficial. We will include more comprehensive derivations in the camera-ready submission.
>
> **Features in Middle Layers / Concrete Examples:** We concur that discussing intermediate features would strengthen the paper. While detailed analysis is ongoing, we will incorporate a discussion of related work [4] on layer-wise feature clustering and plan to add a brief analysis in the Appendix to provide more concrete examples of information captured by intermediate layers.
>
> **Attention Sink** Thank you for highlighting the connection to the attention sink phenomenon [3]. We hypothesize that the mid-layer compression we observe could be related: attention sinks might channel contextual information through a few critical tokens, creating an information bottleneck. We will incorporate a brief discussion of this potential link in the revised manuscript.
>
> **AR vs. Bi-directional:** Regarding the request for a more developed theoretical explanation for differing compression patterns, we acknowledge this is an interesting area. While we attribute the empirical difference to distinct training objectives, a full derivation is complex. We will refine our discussion and note this as an avenue for future theoretical work.
>
> ## Responses to Questions
>
> **1: Best Layers for Different Task Types** We have looked at which layers are optimal for different task types. for Pythia-410M, optimal layers vary: ~50\% depth for Classification, ~75\% for Clustering and Retrieval. This suggests task-dependent preferences. We will include these breakdowns in the Appendix and discuss them.
>
>
>
>  **2: Effect of Model Scale on Optimal Layer Depth** Optimal depth tends to shift deeper with scale (first half for small Pythia models, ~70\% for 160M/410M). This ~70\% depth roughly corresponds to where entropy begins increasing after the dip (Fig 10a, Appendix), suggesting the optimal layer often lies where representational capacity expands post-compression. We will elaborate on this in the revision.
>
>
>
>
> **3: Explicit Optimization During Training**
> Explicit Optimization During Training: While we didn't optimize these metrics during training, we agree it's a promising direction. Concurrent work [2] explored this for finetuning (finding significant compression was less suitable for math tasks requiring full input detail). Optimizing during pre-training is an interesting future direction we will mention.
>
> ## References
>
> [1] Sanyal, et al, "Inheritune: Training smaller yet more attentive language models",  2024
>
> [2] Arefin, et al, "Seq-VCR: Preventing Collapse in Intermediate Transformer Representations for Enhanced Reasoning", 2024
>
> [3] Xiao, et al. "Efficient Streaming Language Models with Attention Sinks." 2024
>
> [4] Chen, et al. "Is Bigger and Deeper Always Better? Probing Llama Across Scales and Layers." 2023

---

### Decision · Program_Chairs · 2025-05-01

**Decision:**

Accept (oral)

**Comment:**

This paper provides a useful perspective on how representations evolve across the layers of (mostly) language models, including across training. The reviewers and I agree that the experiments are thorough, thoughtful, and clearly presented, and that the work helps to elucidate and unify various prior findings. This seems like a valuable contribution to the literature, potentially suitable for a spotlight.